# Structural insights into the regulation of monomeric and dimeric apelin receptor

Yang Yue[1,10], Lier Liu[1,2,10], Lijie Wu [1,10], Chanjuan Xu [3,4,10], Man Na[1,2], Shenhui Liu[1,2], Yuxuan Liu[3,4], Fei Li [1], Junlin Liu[1], Songting Shi[5], Hui Lei[5], Minxuan Zhao[1,2], Tianjie Yang[6], Wei Ji [6], Arthur Wang[7], Michael A. Hanson[8], Raymond C. Stevens[5], Jianfeng Liu [3,4] ✉ & Fei Xu [1,2,7,9] ✉

The apelin receptor (APJR) emerges as a promising drug target for cardio-vascular health and muscle regeneration. While prior research unveiled the structural versatility of APJR in coupling to Gi proteins as a monomer or dimer, the dynamic regulation within the APJR dimer during activation remains poorly understood. In this study, we present the structures of the APJR dimer and monomer complexed with its endogenous ligand apelin-13. In the dimeric structure, apelin-13 binds exclusively to one protomer that is coupled with Gi proteins, revealing a distinct ligand-binding behavior within APJR homo-dimers. Furthermore, binding of an antagonistic antibody induces a more compact dimerization by engaging both protomers. Notably, structural ana-lyses of the APJR dimer complexed with an agonistic antibody, with or without Gi proteins, suggest that G protein coupling may promote the dissociation of the APJR dimer during activation. These findings underscore the intricate interplay between ligands, dimerization, and G protein coupling in regulating APJR signaling pathways.

Accumulating evidence has suggested that G protein-coupled receptors (GPCRs) function not only as monomers but also as dimers/oligomers[1–5]. While extensive and in-depth investigations have been conducted on the dimerization structures of class C and class D GPCRs, there remains a significant research gap in the study of dimerization structure-function relationship within the class A GPCRs. Our previous research findings have unveiled the dimeric structure of the class A GPCR, APJR, which interacts with its down-stream Gi proteins in both 2:1 and 1:1 stoichiometric ratios. The formation of the APJR homodimer occurs through a small, hydro-phobic interface at the junction of transmembrane helix III (TM3) and extracellular loop 1 (ECL1), facilitated by the "FGTFF motif"

(human APJR residues 97-101) (Supplementary Fig. 1a). This dis-tinctive mechanism sets APJR apart from the obligate dimers observed in class C and class D GPCRs. In class C GPCRs, character-ized by their dimeric arrangement and a substantial extracellular domain (ECD) resembling a VFT structure, agonist binding and G protein coupling trigger a closure and reorganization of the VFTs. This action brings the ECD into proximity and leads to a rearrange-ment of the transmembrane domain (TMD) and the dimer interface, facilitating the transition to an active state[6–8]. On the other hand, for the class D GPCR (Ste2), changes in contacting residues and surface area occur during receptor activation while maintaining the integrity of the dimer interface[6]. Despite these insights, the dynamic

[1]iHuman Institute, ShanghaiTech University, Shanghai, China. [2]School of Life Science and Technology, ShanghaiTech University, Shanghai, China. [3]Key Laboratory of Molecular Biophysics of MOE, College of Life Science and Technology, Huazhong University of Science and Technology (HUST), Wuhan, China. [4]International Research Center for Sensory Biology and Technology of MOST, College of Life Science and Technology, Huazhong University of Science and Technology (HUST), Wuhan, China. [5]Structure Therapeutics, South San Francisco, CA, USA. [6]Institute of Biophysics, Chinese Academy of Sciences, Beijing, China. [7]JiKang Therapeutics, Shanghai, China. [8]Phillip and Patricia Frost Institute for Chemistry and Molecular Science, University of Miami, Coral Gables, FL, USA. [9]Shanghai Clinical Research and Trial Center, Shanghai, China. [10]These authors contributed equally: Yang Yue, Lier Liu, Lijie Wu, Chanjuan Xu. ✉e-mail: jfliu@mail.hust.edu.cn; xufei@shanghaitech.edu.cn

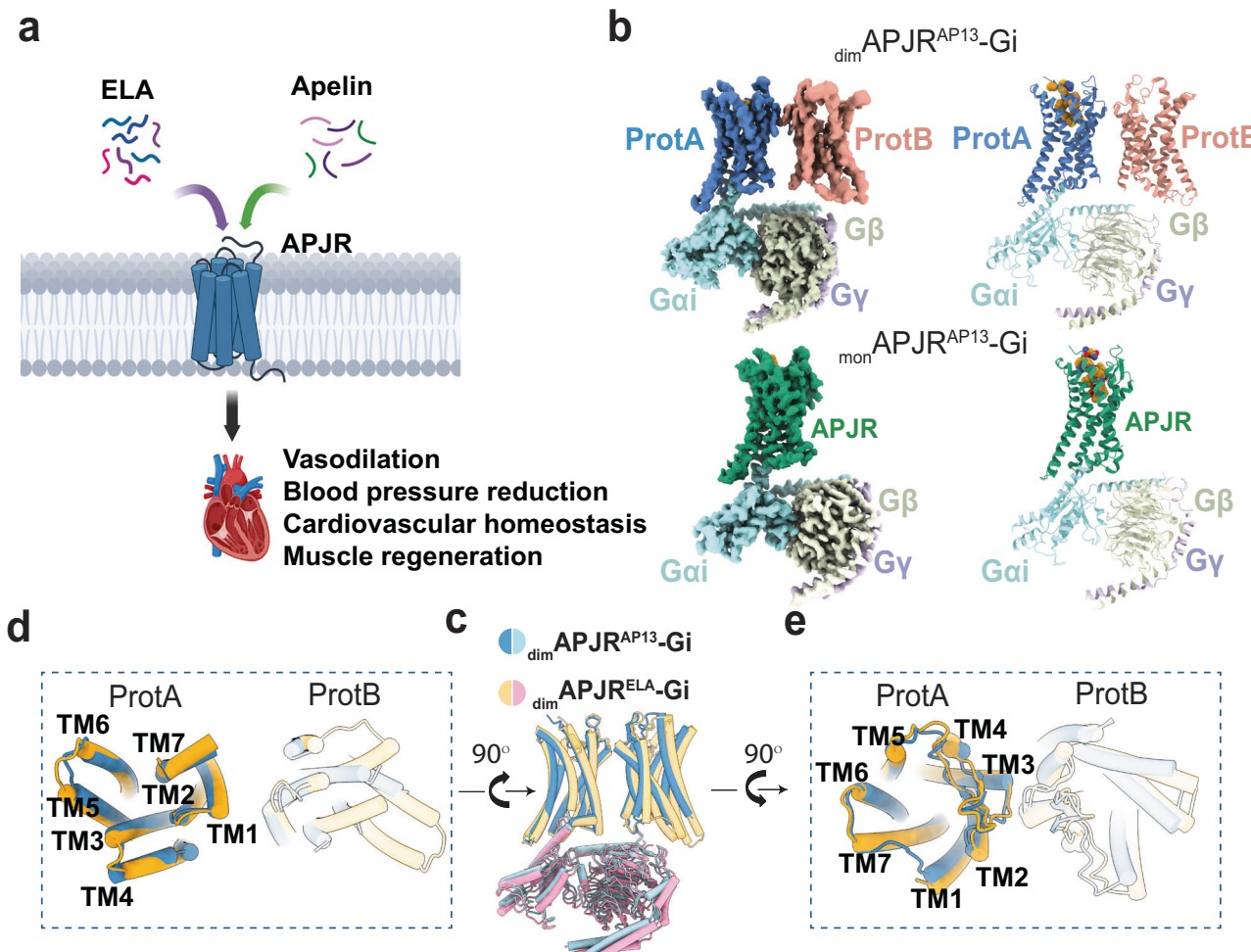

**Fig. 1 | Cryo-EM structures of the dimAPJR^AP13-Gi and monAPJR^AP13-Gi complexes and comparisons with dimAPJR^ELA-Gi complex. a** A model demonstrating the potential pharmacological effects induced by two endogenous peptide ligands on APJR. Created in BioRender. Yue, Y. (2025) https://BioRender.com/p30r650. **b** Cryo-EM maps and structure models of dimAPJR^AP13-Gi complex and monAPJR^AP13-Gi complex. Protomer primarily coupled with Gi is termed ProtA and non-coupled is termed ProtB. Apelin-13, yellow; ProtA^AP13, blue; ProtB^AP13, pink; monAPJR^AP13, green. Gα_i, Gβ and Gγ are in light cyan, light green and light blue, respectively. The ICL1 and ICL2 regions were disordered, and in ProtB, only the main chain was modeled due to insufficient side-chain density. Structural comparison between dimAPJR^AP13-Gi and dimAPJR^ELA-Gi (PDB ID: 7W0N) in overall side view (**c**), intracellular view (**d**), and extracellular view (**e**), respectively.

alterations within the APJR dimer during the activation process remain a subject of continued inquiry.

APJR has been extensively studied as a drug target for heart failure and cardiovascular diseases, with two endogenous peptide ligands: apelin and elabela (ELA)[9-12] (Fig. 1a). Apelin, derived from a 77-amino acid pre-peptide, is cleaved into various isoforms such as apelin-36, apelin-17, and apelin-13, with apelin-13 being the most abundant and exhibiting neuroprotective effects and potent circulatory activity[13-16]. Binding of apelin to APJR induces significant vasodilation and blood pressure reduction[17]. Studies by BioAge Labs, Inc. and others indicate that higher apelin levels correlate with improved physical function and longevity, especially when combined with GLP-1 agonists to enhance weight loss and body composition[18-22]. While our previous work on APJR-apelin analog co-crystal structure and recent research on APJR-apelin-13 cryo-EM structure by Zhang et al. have shed light on how APJR recognizes apelin and its analogs[23,24], the details of APJR dimerization and signaling regulation upon activation by apelin remain to be elucidated.

Through prior structural analyses of the ELA and small molecule cmpd644 bound APJR-Gi complexes, we uncovered the coexistence of both APJR dimers and monomers[25]. In the dimeric structures, the ligands were found in both protomers. To explore whether other ligands exhibit similar behavior, this study initially resolves cryo-electron microscopy (cryo-EM) structure of the APJR-Gi complex with its endogenous peptide agonist, apelin-13. Additionally, to enhance our comprehension of the dynamic changes in the activation process of APJR dimers, we further elucidate the structures of the ligand-free (apo) state, the antagonistic-antibody (JN241[26])-bound state in the absence of Gi proteins, and the agonistic-antibody (JN241-9[26])-bound state of APJR with and without Gi proteins. These structures reveal that apelin-13 binds exclusively to the protomer that is coupled with Gi proteins (herein referred to as ProtA). Moreover, the antagonistic antibody binds to both protomers, inducing a more compact dimerization. Intriguingly, the agonistic antibody induces predominantly dimeric APJR in the absence of Gi proteins but transitions to a monomer-prone state upon Gi proteins binding. This suggests that the coupling of Gi proteins may promote the dissociation of the APJR dimer during activation.

## Results

### Investigating APJR dimerization dynamics at cell surfaces

We initially explored the cell surface dynamics of APJR monomer and dimer formations utilizing single-molecule imaging of Snap-tagged APJR labeled with non-cell-permeant fluorophores, as previously

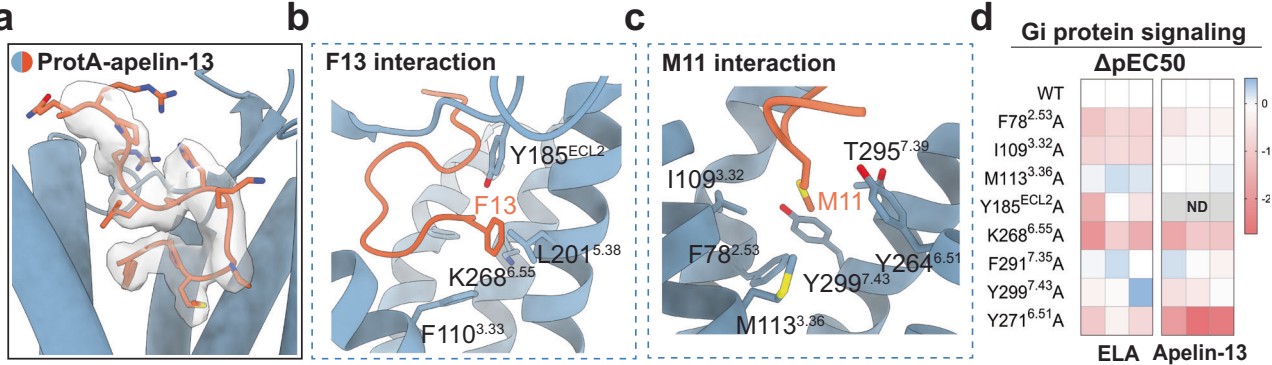

**Fig. 2 | Recognition mechanism of the endogenous apelin-13 by APJR and comparison to ELA binding mode. a** Binding pose of apelin-13. Apelin-13 is shown in orange sticks. ProtA[API3] is shown as blue cartoon. **b, c** Key residues in the apelin-13 binding pocket in APJR. Apelin-13 residues are labeled in orange. Hydrophobic interactions with F13 of apelin-13 (**b**). Residues interacting with M11 of apelin-13 (**c**). **d** Effects of key residue mutations on Gi-protein signaling in the apelin-13 binding pocket of APJR, measured by Glo-Sensor cAMP assay. Heatmap is generated on the basis of the ΔpEC50 (ΔpEC50 = pEC50 of mutant − pEC50 of WT APJR) for either apelin-13 or ELA. Each column represents the data of an independent replicate ($n = 3$). The corresponding data are shown in Supplementary Table 2. "ND" indicates no detectable signal. Source data are provided as a Source Data file.

outlined[1]. The total internal reflection fluorescence microscope (TIRFM), a potent tool for visualizing membrane proteins, facilitates the precise observation of GPCR dimers on the plasma membrane. Individual APJR molecules were visualized using TIRFM, followed by a photobleaching step analysis (Supplementary Fig. 1b). Our findings revealed approximately 16% two-step events (indicative of the dimer population) at a density of $0.48 \pm 0.01$ spots/$\mu m^2$, indicating that APJR dimerization occurs at low receptor concentrations on the cell surface. Upon treatment with cmpd644, a highly potent small molecule targeting APJR[25], a modest increase in the dimer population to approximately 22% was observed, suggesting that the dynamic transitions between the dimeric and monomeric states are intrinsic characteristics of APJR at the cell surface and are minimally influenced by the ligand (Supplementary Fig. 1c). The presence of both monomeric and dimeric states of APJR in living cells aligns with our previous structural findings in the purified system[25]. APJR signaling likely operates through a versatile regulatory mechanism influenced by ligands, G-proteins as well as the oligomerization state—complexities that single-molecule experiments cannot fully resolve. Therefore, we proceeded with a series of structural investigations of APJR complexes in various ligand and G protein conditions to further dissect this regulation.

### Apelin-13-bound APJR-Gi complexes revealed the co-existence of dimer and monomer

Having identified the binding modes of ELA and cmpd644 on the APJR dimer[25], our objective was to delve deeper into the binding interactions of the APJR with another endogenous peptide apelin-13. For the cryo-EM investigations, we co-expressed APJR with Gαi1, Gβ1, and Gγ2 in Trichoplusia ni (Hi5) insect cells and purified the complex in the presence of apelin-13, apyrase, and scFv16, which aided in stabilizing the Gi protein α- and βγ-subunits. Analysis of the cryo-EM data revealed two primary particle classes: one containing dimeric APJR-Gi (dimAPJR-Gi) and the other monomeric APJR-Gi (monAPJR-Gi), similar to the previously reported cmpd644/ELA-activated complex structures (Fig. 1b, c). Subsequent iterative 2D/3D classification, refinement, and model-building steps led to the generation of the apelin-13-dimAPJR-Gi complex (dimAPJR[API3]-Gi) and apelin-13-monAPJR-Gi complex (monAPJR[API3]-Gi) structures, with a global nominal resolution of 3.5 Å and 3.1 Å, respectively (Supplementary Figs. 2, 3 and Supplementary Table 1).

The availability of a series of dimeric structures of APJR bound to agonists provided a unique opportunity to explore the conformational plasticity of APJR homodimers in the presence of different ligands. In all structures, the ProtA subunits exhibited a classically active conformation with the ligands occupying the orthosteric pocket as expected. However, a striking observation was made in the ProtB subunit of the dimAPJR[API3]-Gi complex, where no ligand density was detected in the orthosteric site (Fig. 1b) while the receptor exhibited an inactive-like conformation. This contrasted sharply with the ProtBs in the dimAPJR[cmpd644/ELA]-Gi complexes, where clear ligand densities corresponding to cmpd644/ELA binding were evident[25]. These findings reveal distinct ligand-binding behaviors within the APJR homodimeric structures.

### Molecular recognition of apelin-13 by APJR and comparison to ELA binding mode

We first investigated the ligand-receptor interactions in the monomeric and dimeric APJR structures. Apelin-13 assumes a compact "S shape" as it penetrates deeper into the "site 1" region (as designated in the previous report[24]) (Fig. 2a and Supplementary Fig. 4a). The conserved twelve residues at the C-terminus of apelin isoforms emerge as crucial for receptor activation across species[27], with F13 and M11 orchestrating key interactions within the binding pocket. Our structure unveils the side-chain of F13 in apelin-13 engages a hydrophobic interaction network with surrounding residues F110[3.33], Y185[ECL2] and L201[5.38] (superscripts indicate Ballesteros–Weinstein numbering for GPCRs[28]) (Fig. 2b). Furthermore, M11 delves deep into the binding pocket, mediating the majority of interactions with hydrophobic residues F78[2.53], I109[3.32], M113[3.36], Y264[6.51], T295[7.39] and Y299[7.43] (Fig. 2c), corroborating previous findings on the pivotal role of M11 in receptor function and ligand potency[27]. Although apelin and ELA can both bind to APJR, they are differed in sequences and functions[11] (Supplementary Fig. 4b). Structural comparisons unveil similarities between the structures of ProtA[API3] and ProtA[ELA], with a root mean square deviation (RMSD) of 1.178 Å (Supplementary Fig. 4c). However, notable differences in interaction patterns between these peptides include: 1) apelin-13 tilts towards TM3/4/5, whereas ELA predominantly leans towards TM1 and TM2 (Supplementary Fig. 4d); 2) the residue M11 in apelin-13 penetrates deeper compared to ELA (Supplementary Fig. 4d). Mutations targeting surrounding residues and the cAMP accumulation assays demonstrate that, at comparable expression levels (Supplementary Table 2), the Y185[ECL2]A and Y271[6.58]A mutations markedly decrease the signaling activity of apelin-13, while inducing marginal reduction in potency for ELA in APJR activation (Fig. 2d, Supplementary Fig. 4e and Supplementary Table 2).

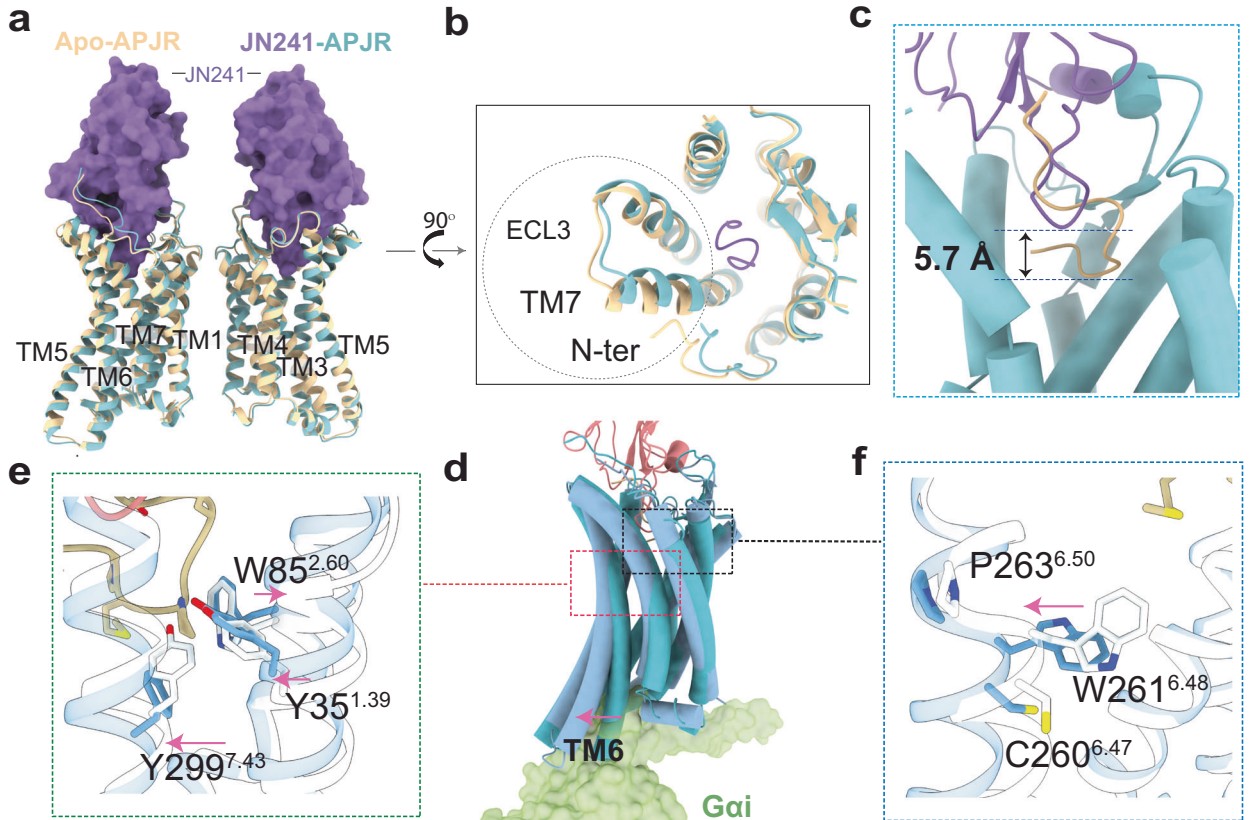

**Fig. 3 | Structural features of antagonist-bound and ligand-free (Apo) APJR structures in dimeric forms and the activation hallmarks.** Structural superposition of JN241-APJR with apo-APJR structures, in overall side view (**a**) and in extracellular view (**b**). The major conformational change between the two APJR structures is circled in dashed lines. Apo-APJR is colored in yellow. APJR in JN241-APJR complex is colored in light cyan, and JN241 is represented as purple surface. **c** Overall superposition of the ProtA[AP13] (apelin-13 in yellow) and one protomer from the JN241-bound symmetric APJR dimer structures (JN241 in purple) to show the deeper insertion of apelin-13 compared to JN241 by 5.7 Å. **d** Overall superposition of the ProtA[AP13] (dark blue) and one protomer from the JN241-bound symmetric APJR dimer structures (light cyan) to highlight the outward movement of TM6 in the apelin-13-bound and Gi-coupled state. **e**, **f** Upon activation, the inward movement of TM1, outward movement of TM2 and TM7 at the extracellular side are indicated as arrows. "Y35[1.39]-W85[2.50]-Y299[7.43]" motif is shown with sticks (**e**). In (**f**) the outward movement of W261[6.48] exhibits a hallmark of APJR activation.

## Structural insights into APJR antagonistic antibody recognition and influence on dimerization

To investigate how the antagonist recognizes APJR and its potential influence on APJR dimerization in the absence of Gi proteins, and to unveil the antibody binding mode which could be beneficial to guide new class of therapeutics development, we utilized a functional single-domain antibody (sdAb) antagonist, JN241, for cryo-EM analysis[26]. The formation of the APJR-JN241-Fc complex was confirmed through size-exclusion chromatography (SEC) and SDS-PAGE analysis (Supplementary Fig. 5a–d). Subsequent cryo-EM analysis of the APJR-JN241-Fc complex (JN241-APJR) revealed a predominant presence of dimeric species, resulting in a density map at a global nominal resolution of 3.0 Å (Supplementary Fig. 6a). Additionally, by processing further datasets of the apelin-13-APJR-Gi complex, we isolated a distinct particle class, enabling the elucidation of the structure of the APJR dimer in its apo state without Gi-protein coupling. This apo-APJR structure, with a global nominal resolution of 3.0 Å (Supplementary Figs. 2, 3 and 6b), displayed an inactive conformation similar to the antagonist-bound APJR structure, with RMSD of 0.604 Å (Fig. 3a). A notable conformational difference between the apo and antagonist-bound APJR structures was observed in the extracellular domain (N-terminus, tips of TM7, and ECL3), likely attributed to the antagonistic sdAb binding (Fig. 3b). Upon antagonist binding, the side-chain of E174[ECL2] from APJR reoriented to form hydrogen bonds with residues T52/R53 from CDR2 and residue C109 from CDR3 in JN241, indicating

the importance of these interactions in JN241 binding (Supplementary Fig. 6c). The structure also highlighted the significant role of all three CDR loops of JN241 in binding to the extracellular side of APJR[26] (Fig. 3a–c). Interestingly, the formation of dimers was not observed in the previously reported JN241-bound APJR crystal structure[26], possibly due to crystallization artifacts and construct modifications. Upon comparing our cryo-EM structure with the crystal structure, we observed that APJR adopts a similarly inactive conformation (Supplementary Fig. 6d). Delving further into the dimer interface, we noted that the interface map density accommodates the five amino acids of the FGTFF motif with high clarity in the cryo-EM map (Supplementary Fig. 6e). Compared to the FGTFF motif in the crystal structure, we detected subtle conformational shifts in the side chains of three pivotal phenylalanine residues. These slight changes are likely due to dimerization effects in the cryo-EM structures.

Intriguingly, in the presence of the antagonist and absence of Gi proteins, nearly no monomers were detected in the 2D classification. To address concerns regarding the impact of the Fc tag on dimer formation, we generated a JN241-APJR complex with the Fc tag removed from the sdAb. 2D classification results consistently demonstrated dimer formation (Supplementary Fig. 6f), although the cryo-EM data quality was insufficient for high-resolution structure determination. Superposition of the active (ProtA[AP13]) and inactive (bound with JN241) dimeric APJR structures provided insights into the molecular details of APJR activation in the dimeric state (Fig. 3d).

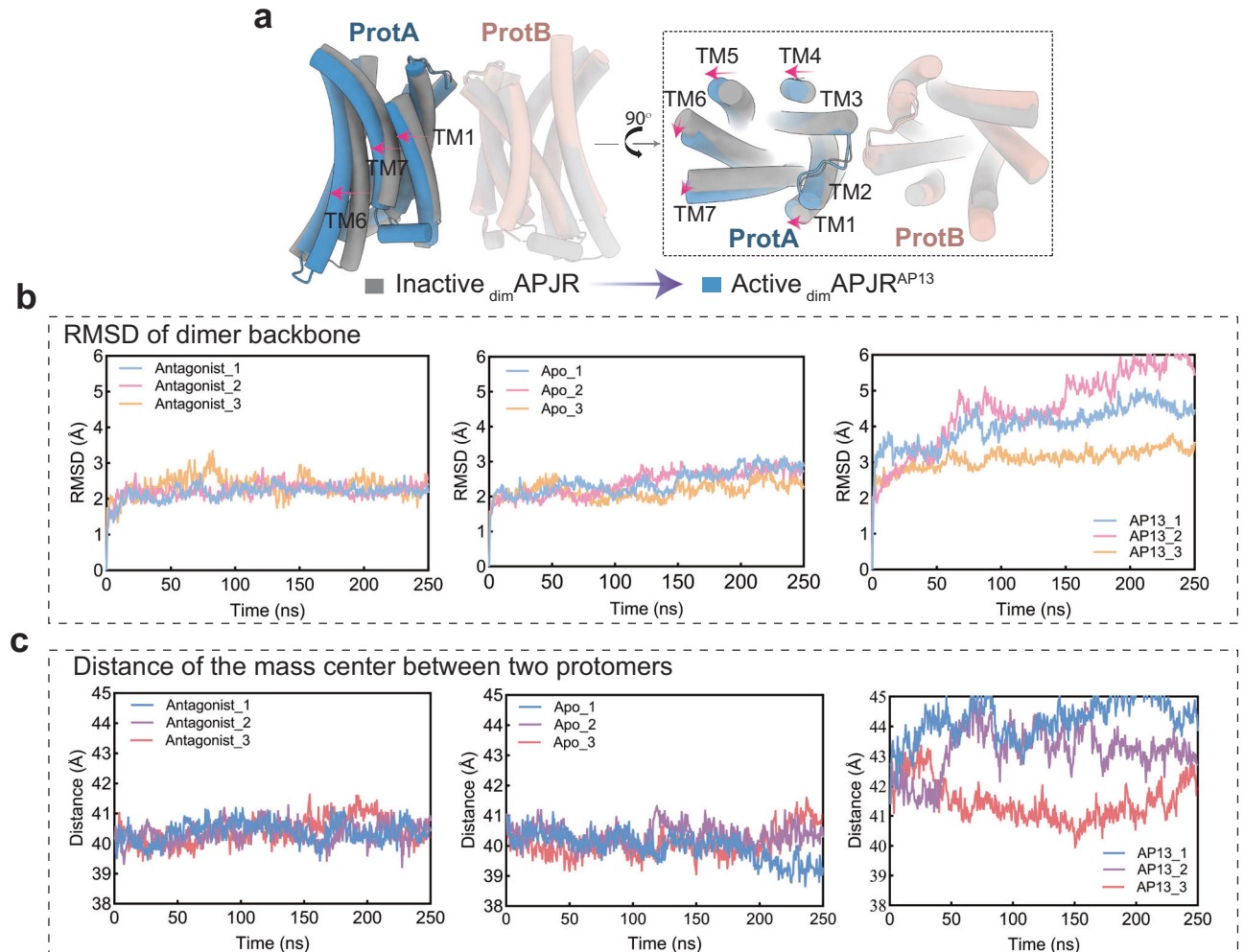

**Fig. 4 | Dimerization-regulated APJR activation features, ligand-dependent ProtB conformation and dimer stability analysis. a** Structural transitions from inactive-state APJR dimer (JN241-bound symmetric dimer) to active-state APJR dimer (agonist-bound asymmetric dimer) upon activation induced by apelin-13. Superposition of $_{dim}$APJR$^{JN241}$ (inactive dimers) (gray) and $_{dim}$APJR$^{AP13}$–Gi (active dimers) (ProtA: blue, ProtB: pink) on ProtB. Gi protein was omitted. The transitional movement related to ProtA from inactive to active states is indicated with red arrows. **b** RMSD of APJR dimer backbone in three different states (apo, inactive and apelin-13-bound) in 250 ns MD simulations repeated in triplicates. **c** Distance of the mass center between two protomers in three different states (apo, inactive and apelin-13 bound) in 250 ns MD simulations repeated in triplicates.

Notably, apelin-13 exhibited a deeper insertion compared to JN241 (Fig. 3c), leading to favorable interactions with key motif "Y35$^{1.39}$-W85$^{2.50}$-Y299$^{7.43}$", which is conserved in angiotensin II receptor type I (AT1R) and angiotensin II receptor type II (AT2R)[29,30], resulting in specific conformational changes in TM1, TM2, and TM7 at the extracellular side (Fig. 3e). Particularly, the interaction of the agonist induced an essential outward movement of the "toggle switch" W261$^{6.48}$, facilitating the formation of the Gi-protein binding cleft during receptor activation (Fig. 3f, and Supplementary Fig. 6g).

**Inactive-to-active structural analysis revealed potential APJR activation mechanism modulated by G-protein binding and dimer dissociation**

In our exploration of the APJR dimer, ranging from an inactive state to an active state bound to various agonists, a thorough investigation into the dynamic changes occurring within APJR dimer during this transition is crucial. Comparing the structures of agonist-bound dimers with the antagonist-bound counterpart, we first made intriguing observations of cholesterol molecules at the dimer interface, potentially contributing to dimer stabilization (Supplementary Fig. 6a, b). The interfacial contacts formed by the "FGTFF motif"

residues displayed consistent configurations across all structures, emphasizing their role in dimer stability (Supplementary Fig. 7a). Nevertheless, we observed a relative shift (measured at approximately 3.2 Å on the Cα of Y299) in the helical bundle of ProtA away from ProtB during the transition to the Gi-protein coupled active state (Fig. 4a). Furthermore, comparing monomeric and dimeric APJR-Gi complexes revealed a lateral shift in the monomeric APJR at the extracellular surface, while the intracellular surface remained relatively unchanged, likely due to the constraints imposed by Gi-protein coupling (Supplementary Fig. 7b).

These structural insights led us to hypothesize a dynamic transformation of the dimer to a monomer during Gi-protein coupling. Molecular dynamics (MD) simulations supported this hypothesis, showing that the dimer state in apo or antagonist-bound structures is more stable than in agonist-bound and Gi-protein coupled structures, as indicated by the RMSD of the overall backbone of APJR dimers (Fig. 4b). Additionally, the distance between the two protomers increases during agonist-induced activation in the presence of Gi protein (Fig. 4c), suggesting a potential mechanism that destabilization of dimerization leads to a transition from dimer to monomer upon agonist binding and G-protein coupling.

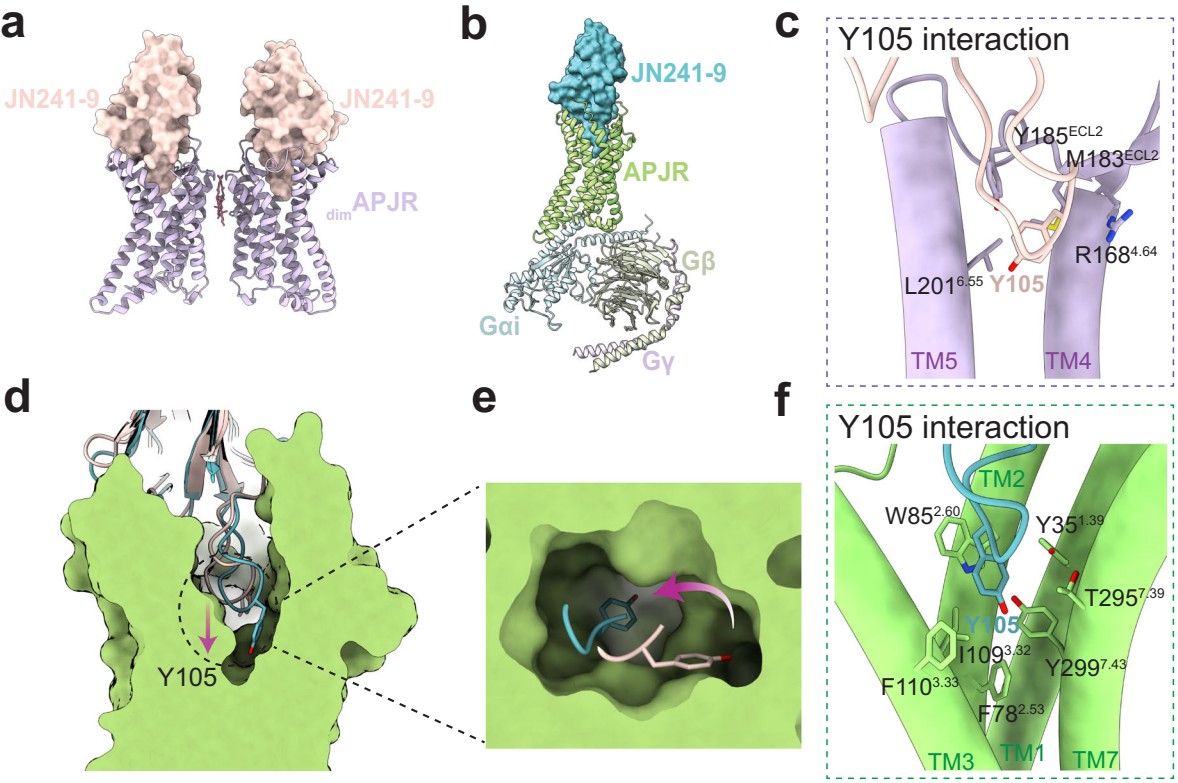

**Fig. 5 | Agonistic antibody binding modes in the absence and presence of Gi proteins. a** Structural model of the dimeric JN241-9-APJR complex. APJR is depicted in purple; JN241-9 is shown in light pink surface. **b** Structural model of the monomeric JN241-9-APJR-Gi complex. APJR is illustrated in green; JN241-9 is represented in blue surface; Gαi, Gβ, and Gγ are colored in light cyan, light green, and light blue, respectively. **c** Interactions between the Y105 residue in JN241-9 and APJR in the absence of Gi proteins. The interacting residues are shown as sticks. Structural comparison of JN241-9 within the JN241-9-APJR and JN241-9-APJR-Gi complexes shown in side view (**d**) and top view (**e**). The conformational rotation of Y105 during G protein coupling is indicated with arrows. **f** Interactions between the Y105 residue in JN241-9 and APJR in the presence of Gi proteins. The interacting residues are shown as sticks.

## Structural investigation of APJR bound to agonistic antibody in the presence or absence of G proteins

However, these MD simulation results could not conclusively support the role of G-protein in facilitating the dissociation of the APJR dimer. Thus, to further validate this hypothesis, we solved the cryo-EM structures of the agonistic antibody JN241-9 bound APJR in the presence or absence of Gi proteins and conducted comprehensive structural analysis (Fig. 5a, b and Supplementary Fig. 4e–l). JN241-9, derived from JN241 by introducing a tyrosine residue (Y105) into the CDR3 region, exhibits full-agonist properties[26]. The structural findings reveal that in the absence of Gi proteins, the agonistic antibody bound to APJR remains in an inactive state and predominantly exists in a dimeric state (Supplementary Fig. 8a). However, upon co-expression of Gi proteins, a remarkable transition of APJR to an active state occurs, along with the significantly reduced dimeric species (Fig. 5b and Supplementary Fig. 8a). This compelling observation lends further credence to our hypothesis that the introduction of G proteins can trigger the dissociation of dimers, with the extent of dissociation being modulated by the specific ligand.

By comparing the G proteins in JN241-9-APJR-Gi and _dim_APJR^AP13-Gi complexes, we observed that, in the JN241-9-APJR-Gi structure, the absence of ProtB results in a substantial relocation of the Gα and Gβγ subunits towards the region that would otherwise be occupied by ProtB in the _dim_APJR^AP13-Gi complex (Supplementary Fig. 8b). We also performed the structural comparison between JN241-9-APJR-Gi complex and the monomeric mutant F101A^ELA-APJR-Gi structure that we have previously described[1] (PDB ID: 7W0P). The conformation of the receptor largely mirrors that observed in the prior structure. However,

substantial rotations were observed with the Gα and Gβγ subunits (Supplementary Fig. 8c). This rotation suggests that different ligands may induce distinct conformational changes in the G proteins, highlighting the potential for ligand-specific receptor activation profiles.

Moreover, comprehensive structural analysis of the agonistic antibody binding modes in both scenarios unraveled intriguing insights. Although in the absence of Gi proteins, the agonist antibody could effectively bind to APJR, with the pivotal residue Y105 snugly nestled in the crevice between TM4/5 and ECL2 of APJR (Fig. 5c). In contrast, in the presence of Gi proteins, a notable shift in the antibody's binding mode occurs, allowing for a deeper penetration into the receptor's orthosteric binding pocket (Fig. 5d). The conformational rotation of Y105 towards TM2/3 and TM6/7 in this context might play a crucial role in triggering receptor activation (Fig. 5e, f and Supplementary Fig. 8d, e). This dynamic alteration not only highlights the remarkable plasticity and adaptability of the APJR ligand binding pocket but also provides structural evidence supporting the notion that the incorporation of Gi proteins enables the ligand to bind even more profoundly within the pocket. The activation of GPCRs is commonly acknowledged to necessitate the engagement of agonists and G proteins. While certain GPCRs can achieve full or partial activation through the sole action of agonists[31,32], others necessitate G protein binding to transition into an active state[33,34]. Our structural analyses indicate that, in the case of APJR, agonist binding alone may not be capable of achieving the full activation as evidenced by the conformations of the receptors we have observed before[24,25] as well as in this study: the TM6 lacks outward movement typically seen in class-A GPCR activation (Supplementary Fig. 8a). These findings underline the critical role of G proteins in facilitating APJR activation.

## Discussion

In this study, we uncovered significant disparities in the binding patterns of various ligands to the APJR dimer. Moreover, dynamic interaction between the two protomers may impose an allosteric influence on signal integration within the dimer. We inferred that in the absence of Gi-protein binding, the agonist may interact with both orthosteric sites within the dimer. Upon Gi protein coupling, ProtA, now complexed with the Gi protein, effectively stabilizes the ligand at its binding site. This aligns with the mechanism through which G proteins mediate the enhancement of agonist activity[35,36]. In contrast, ProtB, without G-protein interaction, potentially suffers from reduced ligand stability in its binding pocket, potentially leading to ligand release and rendering ProtB inactive. For ligands with limited binding affinities with APJR, or those as flexible as apelin-13, ProtAs might exert an inhibitory effect on the ligand binding and activation of ProtBs. Conversely, ligands with higher affinity (such as cmpd644, PDB ID: 7WOL[25]), or those with extensive interactions, (such as ELA, PDB ID: 7WON[25]), seem to have minimal impact as they can still bind to ProtBs. While this hypothesis requires further exploration, it offers a plausible mechanism explaining how G-protein binding enhances agonist binding to the receptors (ProtAs) whereas ProtB may function as a negative allosteric modulator, reducing the overall signaling output when forming the dimer with ProtA. Furthermore, in certain instances, like with the gamma-aminobutyric acid receptor (GABAB)[37] and gustatory receptors[38], heterodimerization is essential for receptor functionality. The potential formation of heterodimers between APJR and other GPCRs raises intriguing questions regarding the modulation of APJR ligand binding and Gi-protein coupling by these GPCRs, warranting further investigation. For class A GPCR heterodimers, a plausible hypothesis is that one protomer can influence the function of the other through agonistic or antagonistic mechanisms, with specific ligands inducing cooperative functional selectivity. For instance, previous studies have demonstrated that apelin can stimulate APJR to form a heterodimer with AT1R, decreasing the affinity of angiotensin II (endogenous ligand of AT1R) and inhibiting its signaling[39]. Similar negative allosteric modulation has been observed between adenosine A1 receptors (A1R) and dopamine D1 receptors (D1R) in the brain, where coactivation of these receptors enhances heteromerization and reduces cAMP accumulation compared to D1R activation alone[40,41]. These findings may inspire future research into the structure-function relationship of GPCR homo- and hetero-dimerization, with our current structural studies laying the foundation for further exploration in this area.

Although the dynamic changes of class C and class D GPCR dimers during G protein coupling have been investigated, the impact of G proteins on class A GPCR dimers remains unknown. Our investigation of the APJR dimer, encompassing its transition from an inactive to an active state bound to various agonists, revealed substantial dynamic regulations. Comprehensive structural analysis revealed that in the absence of G protein interaction, whether bound to an antagonist, agonist, or in the apo state, the APJR forms compact and nearly exclusive dimers. Moreover, the dimer interface remains largely unchanged (Supplementary Fig. 9). However, this characteristic sets it apart from class C GPCRs[42,43]. Notably, the binding of agonists to class C GPCRs triggers a relative shift between the two subunits. This shift is facilitated by helix VI in both subunits, a mechanism widely recognized as a key feature of class C receptor activation[42,43]. Subsequently, upon binding to G proteins, the distance between the two protomers of APJR increases (Fig. 4a), indicating dimer destabilization upon agonist binding and G protein coupling. Further validation of this hypothesis through the examination of structural interactions between the full-agonist antibody JN241-9 and APJR elucidates the pivotal role of G proteins in promoting dimer dissociation and subsequent receptor activation. These findings provide further insights into the mechanistic role of G proteins in class A GPCR dimer dynamics and receptor activation.

Finally, our comprehensive analysis of the structure-function relationship has yielded valuable insights into the dimerization mechanism of APJR, unveiling substantial differences in the binding modes of diverse ligands to the APJR dimeric form. Additionally, our findings highlight the critical role of G proteins in modulating the interaction between protomers within the GPCR dimer and regulating the receptor activation. Given the imperative medical need to target APJR for agonist development, these insights not only offer valuable strategies for the discovery and design of drugs targeting the APJR dimer but also pave the way for further research directions and methodologies in the fields of GPCR oligomerization and drug development.

## Methods

### Cloning and co-expression of APJR and Gi heterotrimer for cryo-EM study

The gene *APLNR*, encoding human WT APJR, was sub-cloned into the pFastBac1 vector with a deletion of 50 residues from the C-terminus, preserving the wild-type sequence upstream. An N-terminal fusion consisting of a Haemagglutinin (HA) signal peptide, Flag tag, 10x His tag, and BRIL fusion[44] was introduced. Separately, human Gαi1 with three dominant-negative mutations (S47N, G203A, A326S)[45] and Gβ1γ2 were cloned into pFastBac1 and pFastBacDual vectors, respectively. For APJR[AP13]-Gi complex, human APJR, Gαi1 with three dominant-negative mutations, Gβ1, and Gγ2 were co-expressed in Trichuplusia ni (Hi5) insect cells using the Bac-to-bac system. The cells were infected with baculoviruses for APJR, Gαi1 with three dominant-negative mutations, and Gβ1γ2 at a 1:3:2 ratio at a density of $2 \times 10^6$ cells per mL. For APJR expression alone, cells were infected with the APJR baculovirus at the same density. After 48 h of culture at 27 °C post-infection, cells were collected by centrifugation and the cell pellets were stored at −80 °C. For cryo-EM studies of the APJR-JN241-9 and APJR-JN241-9-Gi complexes, APJR and JN241-9 were co-expressed in Trichuplusia ni (Hi5) insect cells using the Bac-to-Bac system. The cells were infected with baculoviruses for APJR and JN241-9 at a 1:1 ratio, or for APJR, JN241-9, Gαi1, and Gβ1γ2 at a 1:1:3:2 ratio, at a density of $2 \times 10^6$ cells per mL.

### Expression and purification of scFv16

The purification of scFv16 has been reported before and we purified it in the similar way[46]. scFv16, featuring an 8x His tag, was sub-cloned into the pFastBac1 vector. The protein was expressed in Trichuplusia ni (Hi5) insect cells as a secreted product and subsequently purified using Ni-NTA affinity chromatography. The process entailed adjusting the pH of the supernatant to 8.0 with 1 M Tris, followed by a 2 h incubation with Ni-NTA resin at 4 °C. The resin was then transferred to a gravity column and sequentially washed with 6 column volumes (CV) of wash buffer I (20 mM HEPES (ABCONE), pH 7.5, containing 100 mM NaCl and 10 mM imidazole) and 4 CV of wash buffer II (20 mM HEPES, pH 7.5, with 100 mM NaCl and 30 mM imidazole). Finally, scFv16 was eluted using Elute buffer (20 mM HEPES, pH 7.5, 100 mM NaCl, and 250 mM imidazole).

### Purification and formation of APJR-Gi-scFv16 complex

Cell pellets from 1 L of APJR-Gi co-expression culture were resuspended in 120 mL of hypotonic buffer containing 10 mM HEPES pH 7.5, 10 mM MgCl₂, 20 mM KCl, and a protease inhibitor cocktail (Roche). To facilitate complex formation, 20 μM apelin-13 (GenScript, Nanjing, China) and 0.25U Apyrase (Sigma, 0.5U/μL) were added into the suspension, followed by an overnight incubation at 4 °C. Post-centrifugation at $140,000 \times g$ for 20 minutes at 4 °C, the resulting pellet was resuspended in 18 mL of the same hypotonic buffer. Solubilization was achieved by mixing the suspension with an equal volume of solubilization buffer, which consisted of 100 mM HEPES pH 7.5, 200 mM NaCl, 1% (w/v) LMNG (Anatrace), and 0.2% (w/v) CHS (Sigma),

supplemented with 20 μM apelin-13 and 0.08U Apyrase, and incubating for 2 hours at 4 °C. The solubilized supernatant was collected after another centrifugation step under the same conditions. Subsequently, 50 μL of Talon superflow metal affinity resin (Clontech) and 20 mM imidazole were added to the solubilized supernatant, followed by an overnight incubation at 4 °C. The mixture was then loaded onto an Econo-Pac disposable chromatography column and washed with 14 CV of wash buffer composed of 50 mM HEPES pH 7.5, 100 mM NaCl, 5% glycerol, 0.01% (w/v) LMNG, 0.002% (w/v) CHS, 30 mM imidazole, and 20 μM apelin-13. Elution of the complex was performed with 4 CV of elution buffer containing 50 mM HEPES pH 7.5, 100 mM NaCl, 5% glycerol, 0.01% (w/v) LMNG, 0.002% (w/v) CHS, 200 mM imidazole, and 20 μM apelin-13. The eluted protein was then combined with 125 μg of scFv16 and incubated for a further hour at 4 °C. This mixture underwent size-exclusion chromatography on a Superdex 200 10/300 GL column (GE Healthcare) pre-equilibrated with a buffer of 20 mM HEPES pH 7.5, 100 mM NaCl, 0.001% LMNG (w/v), 0.0002% CHS (w/v), and 1 μM apelin-13. The peak fractions were analyzed by SDS-PAGE and concentrated to a final protein concentration of 1 mg/mL. For the study of JN241-9-APJR-Gi complex, no ligand was added during the complex formation and purification process due to the co-expression of JN241-9, APJR and Gi. All other procedures were carried out as described above.

## Purification and formation of JN241-APJR and JN241-9-APJR complexes

Pellets harvested from 1 L of cultured cells were resuspended in a hypotonic buffer (10 mM HEPES pH 7.5, 10 mM MgCl$_2$, 20 mM KCl), and a protease inhibitor cocktail. Subsequent washing was performed using a hypertonic buffer (10 mM HEPES pH 7.5, 10 mM MgCl$_2$, 20 mM KCl, 1 M NaCl, protease inhibitor cocktail) in a 100 mL Kimble™ Kontes™ Dounce homogenizer. Post-centrifugation at 140,000xg for 40 min at 4 °C, the receptor-containing pellet was resuspended in the initial hypotonic buffer. To extract the receptor, a solubilization buffer (100 mM HEPES pH 7.5, 1.6 M NaCl, 2% (w/v) LMNG (Anatrace), 0.4% (w/v) CHS) was added in a volume equal to that of the suspension, and the mixture was incubated at 4 °C for 2 h. Following a subsequent centrifugation step to clear the mixture, the supernatant was combined with Talon superflow metal affinity resin and 20 mM imidazole, then incubated overnight at 4 °C. The resin was packed into a gravity column and washed with 14 CV of wash buffer containing 25 mM HEPES, 500 mM NaCl, 5% (w/v) glycerol, 0.05% (w/v) LMNG, 0.01% (w/v) CHS, 30 mM imidazole, at pH 7.5. Elution was carried out with 4 CV of elution buffer (25 mM HEPES, 500 mM NaCl, 5% (w/v) glycerol, 0.01% (w/v) LMNG, 0.002% (w/v) CHS, 200 mM imidazole, pH 7.5). To this elute, 125 μg of JN241 (provided by Structure Therapeutics) was added, and the mixture was incubated for an additional hour at 4 °C. The resulting mixture was then subjected to size-exclusion chromatography using a Superdex 200 10/300 GL column (GE Healthcare) equilibrated with a buffer of 20 mM HEPES pH 7.5, 100 mM NaCl, 0.001% LMNG, and 0.0002% (w/v) CHS. Peak fractions collected from this chromatography were analyzed via SDS-PAGE and concentrated to a final protein concentration of 1.5 mg/mL. For the study of the JN241-9-APJR complex, the co-expression method was utilized, and no ligand was added during the purification process. All other procedures were conducted as described above.

## Cryo-EM sample preparation and data collection

Holey carbon grid (CryoMatrix Amorphous alloy film R1.2/1.3, 300 mesh) was glow-discharged for 45 s with H$_2$/O$_2$. 3 μL complex was then applied on the grid, using the Vitrobot Mark IV (Thermo Fisher Scientific, USA) to prepare the sample. The chamber of Vitrobot was set to 100% humidity, 4 °C and the sample preparation parameters were set to blot time 3 s with blot force −1. The cryo-EM dataset was collected on a Titan Krios 300 kV electron microscope (Thermo Fisher Scientific, USA). The calibrated magnification was 165,000 with the pixel size of 0.832 Å/pixel for APJR$^{AP13}$-Gi, JN241-APJR, JN241-9-APJR and JN241-9-APJR-Gi complex. Each movie consisted of 40 frames with a total dose of 60 e$^-$/Å$^2$, and the dose rate was 15 e$^-$/Å$^2$/s. Data collection was done using SerialEM v3.8.0 software with a defocus range of −0.7 μm to −2.2 μm.

## Cryo-EM image processing and 3D reconstruction

For the structural analysis of the APJR$^{AP13}$-Gi complex and apo-APJR, a dataset comprising 9,108 movies was captured and subsequently processed using cryoSPARC v3.0 software[47]. Beam-induced motion artifacts were corrected by applying the patch motion correction algorithm. The Contrast Transfer Function (CTF) parameters for each dose-weighted micrograph were determined using the patch CTF estimation module. Following Auto blob picking, a total of 5,304,464 particles were extracted. These particles underwent 2D classification, resulting in the selection of 980,091 particles for the generation of initial models. These models served as a basis for further 3D classification through heterogeneous refinement within cryoSPARC. Three distinct particle populations were discerned, corresponding to a dimeric APJR complex, a monomeric APJR complex, and the apo-APJR state. A subsequent round of 3D classification yielded refined particle subsets: 24,309 particles for the dimeric APJR complex, 40,681 particles for the monomeric APJR complex, and 224,827 particles for the apo-APJR complex. These subsets were subjected to final homogeneous refinement, non-uniform refinement, and local refinement in cryoSPARC, culminating in density maps with nominal resolutions of 3.48 Å for the $_{dim}$APJR$^{AP13}$-Gi complex, 3.13 Å for the $_{mon}$APJR$^{AP13}$-Gi complex, and 2.97 Å for apo-APJR. The resolutions were established based on the gold standard Fourier shell correlation (FSC) at the 0.143 threshold. Local resolution variations were assessed using the local resolution estimation tool in cryoSPARC. To enhance the local density of maps, automatic masking and local sharpening procedures were conducted utilizing DeepEMhancer[48].

For the structural analysis of the JN241-APJR-Gi complex, a dataset comprising 3,115 movies was captured and subsequently processed using cryoSPARC v3.0 software[47]. Beam-induced motion artifacts were corrected by applying the patch motion correction algorithm. The Contrast Transfer Function (CTF) parameters for each dose-weighted micrograph were determined using the patch CTF estimation module. Following Auto blob picking, a total of 3,071,733 particles were extracted. These particles underwent 2D classification, resulting in the selection of 510,362 particles for the generation of initial models. After several rounds of 3D classification, 170,746 particles were selected out for final homogeneous refinement followed by non-uniform refinement and local refinement in cryoSPARC, culminating in density maps with nominal resolutions of 2.95 Å for the JN241-APJR. The resolutions were established based on the gold standard Fourier shell correlation (FSC) at the 0.143 threshold. Local resolution variations were assessed using the local resolution estimation tool in cryoSPARC. To enhance the local density of maps, automatic masking and local sharpening procedures were conducted utilizing DeepEMhancer[48].

For the structural analysis of the JN241-9-APJR complex, a dataset comprising 4886 movies was captured and subsequently processed using cryoSPARC v3.0 software[47]. Beam-induced motion artifacts were corrected by applying the patch motion correction algorithm. The Contrast Transfer Function (CTF) parameters for each dose-weighted micrograph were determined using the patch CTF estimation module. Following Auto blob picking, a total of 2,203,592 particles were extracted. These particles underwent 2D classification, resulting in the selection of 488,065 particles for the generation of initial models, final homogeneous refinement followed by non-uniform refinement and local refinement in cryoSPARC, culminating in density maps with nominal resolutions of 3.01 Å for the JN241-9-APJR. The resolutions were established based on the gold standard Fourier shell correlation

(FSC) at the 0.143 threshold. Local resolution variations were assessed using the local resolution estimation tool in cryoSPARC. To enhance the local density of maps, automatic masking and local sharpening procedures were conducted utilizing DeepEMhance.

For the structural analysis of the JN241-9-APJR-Gi complex, a dataset comprising 4,071 movies was captured and subsequently processed using cryoSPARC v3.0 software[47]. Beam-induced motion artifacts were corrected by applying the patch motion correction algorithm. The Contrast Transfer Function (CTF) parameters for each dose-weighted micrograph were determined using the patch CTF estimation module. Following Auto blob picking, a total of 1,047,133 particles were extracted. These particles underwent 2D classification, resulting in the selection of 441,368 particles for the generation of initial models. After several rounds of 3D classification, 65,206 particles were selected out for final homogeneous refinement followed by non-uniform refinement and local refinement in cryoSPARC, culminating in density maps with nominal resolutions of 3.12 Å for the JN241-9-APJR-Gi complex. The resolutions were established based on the gold standard Fourier shell correlation (FSC) at the 0.143 threshold. Local resolution variations were assessed using the local resolution estimation tool in cryoSPARC. To enhance the local density of maps, automatic masking and local sharpening procedures were conducted utilizing DeepEMhance.

### Cryo-EM model building and refinement

Reference models with Protein Data Bank (PDB) identifiers 7W0L and 7W0M were utilized for model construction and iterative refinement against the electron density map. Components of the target models were initially positioned within the electron microscopy density map employing UCSF Chimera v1.15[49], succeeded by manual modifications and iterative rebuilding via Coot v0.8.9[50], and subsequent real-space refinement using Phenix v1.14[51]. Model quality was assessed and validated by MolProbity 4.2[52]. Visualization and preparation of structural figures were achieved with UCSF Chimera, Chimera X v1.2.4, and PyMOL v2.5.1 (http://www.pymol.org). Comprehensive refinement metrics are detailed in Supplementary Table 1.

### Single-molecule photobleaching analysis

Single-molecule photobleaching analysis was performed as previous reported[1]. In brief, COS-7 cells were transfected with the SNAP-tagged wild-type APJR using Lipofectamine 2000 and plated in a 35 mm confocal dish. After 24 h transfection, cells were labeled with SNAP-Surface Alexa Fluor® 647 (1 μM) (NEB cat. no. S9136S) for 30 min and fixed with 4% paraformaldehyde for 15 min at RT. Then, after three times of washing with PBS, Trolox (2 mM) was added in the cell to avoid dye blinking and images were taken[53].

A total internal reflection fluorescence (TIRF) microscope, equipped with a high-NA TIRF objective (Olympus Oil ×100, NA = 1.45) and an electron-multiplying charge-coupled device (AndoriXon DV-897 BV), was adopted to achieve single-molecule detection, and a solid-state 650 nm laser (OPSL, Coherent) with 1 mW was used. To avoid photobleaching before image acquisition, cells were searched and focused in a bright field, and a fine focus adjustment in TIRF mode was performed using only 2 % laser power. This procedure results in negligible photobleaching. Afterwards, the laser power was then set to 40 % and an recording images every 50 ms, for a total of 75 s of image sequence (1500 frames). Sequences of images were stored directly on a computer hard drive for subsequent analysis.

Imaging data were analyzed by a home-written MATLAB (Version 2017a) code. Briefly, in order to record the complete photobleaching step, the recording was made earlier than the time the laser was switched on. First the total brightness of each frame was counted, the number of frames in which the laser was switched on will be marked, and the next 1000 frames were intercepted. All light points were extracted for the first frame and tracked, light intensity was counted

and plotted, the bleaching steps were determined manually by one investigator and recorded blindly by another.

Based on the labeling efficiency of 90% in our previous work using the same SNAP-647 dye[54], here we use 0.9 as a correction factor for the binding between dye and protein units.

### Split luciferase biosensor cAMP accumulation assay

The wild-type and mutant *APLNR* gene was cloned into the pcDNA3.1 vector, incorporating an N-terminal hemagglutinin (HA) signal sequence and a Flag epitope for subsequent detection. HEK293T cells (ATCC, CRL-11268) were propagated in Dulbecco's Modified Eagle Medium (1x DMEM, Life Technologies) enriched with 10% fetal bovine serum (FBS) and maintained in a controlled atmosphere of 5% $CO_2$ at 37 °C. To assess the impact of QuickChange PCR-derived mutations on the APJR-Gi protein signaling pathway, a split luciferase GloSensor cAMP biosensor assay (Promega) was employed. 24 h before conducting the assay, HEK293T cells were co-transfected with 1 μg of APJR DNA alongside 1 μg of pGloSensor™-22FGloSensor cAMP DNA (Promega) using Lipofectamine 2000 (Life Technologies) in 6 cm dish. Following a 24 h culture period, cells were seeded into poly-L-lysine-coated 384-well white assay plates (Greiner) at a density of 10,000–15,000 cells within 40 μL of medium per well and incubated overnight (16-20 hours). Subsequent experiments commenced with the addition of 20 μL of 2 mg/mL D-luciferin sodium salt solution in Hanks' Balanced Salt Solution (HBSS, pH 7.4) to each well, followed by a 1 h incubation at 37 °C. Agonist-induced and constitutive activities were evaluated by introducing 10 μL of agonists in buffer to the wells, reaching final concentrations in the range of 0 to 1 30 μM, and incubating for 15 min at 37 °C. To measure agonist activity, wells received an additional 10 μL of isoproterenol (Sigma) to achieve a final concentration of 200 nM, followed by a 15 to 20 min incubation at 37 °C. The intracellular cAMP levels were quantified using an EnVision multiplate reader (Perkin Elmer) and data were processed with GraphPad Prism software version 9.0. Cell surface expression of wild-type and mutant APJR was quantified using fluorescence-activated cell sorting (FACS). Briefly, cells expressing APJR were incubated with an anti-Flag M2–fluorescein isothiocyanate (FITC)-conjugated antibody (Sigma) for 20 min at 4 °C. The cells were subsequently washed with HBSS, and the surface expression levels of APJR were determined by measuring the FITC fluorescence intensity via a Guava EasyCyte HT flow cytometer (Millipore).

### Molecular dynamics simulation of APJR

In the system of the APJR-Gi complex, scFv16 was excluded while preserving the remaining components. The missing intracellular loop 2 (ICL2) of ProtB in the $_{dim}$APJR$^{AP13}$-Gi complex was reconstructed by utilizing the corresponding symmetrical ICL2 from ProtA of the same complex. Initial receptor preparation, including hydrogen addition, terminal capping, and verification of protonation states of titratable residues, was facilitated by Schrödinger software[55]. Notably, residues $D^{2.50}$ and $D^{3.49}$ were manually protonated to reflect the transient protonation states that occur during GPCR activation[56]. The prepared structures of the three complexes (aplein-13-APJR-Gi, JN241-APJR and apo-APJR) were each embedded into a lipid bilayer composed of 405 POPC molecules, utilizing the CHARMM-GUI membrane builder for setup[57]. The positioning of the receptor within the membrane was informed by data from the OPM database[58]. Subsequent solvation of each receptor-membrane configuration occurred in a TIP3P periodic water box with 0.15 M NaCl, ensuring at least a 15.0 Å buffer of water molecules surrounding the bilayer. Ligand parameters were derived using the CGenFF tool within the CHARMM general force field[59].

The molecular dynamics simulations were executed with GROMACS 2020, leveraging the CHARMM36m all-atom force field[60,61]. Parameter files for the simulations were sourced from the CHARMM-

GUI database. The systems were initially subjected to a minimization process. Following this, leveraging the outcomes from the energy minimization, we independently reiterated all subsequent steps. A gradual heating process from 0 to 310 K in the NVT ensemble over a span of 300 ps was conducted, using a time step of 1 fs. Next, equilibration was executed in the NPT ensemble (310 K, 1 atm semi-isotropic) for 10 ns, gradually reducing positional restraints on protein, ligand, and lipid atoms. Post-equilibration, production simulations were conducted for 250 ns with an integration time step of 2 fs for each system. Constraints on bonds involving hydrogen atoms were established using the LINCS algorithm[62]. Electrostatic interactions were computed employing the particle mesh Ewald method[63] with a 12 Å cutoff. Trajectory analyses were performed with the aid of VMD[64] and native GROMACS utilities, with trajectory snapshots captured every 500 ps. GROMACS *mdp input file for the production run is available as Supplementary Data 1, and representative snapshots after 250 ns production run for antagonist bound APJR, apo APJR and AP13 bound APJR are available as Supplementary Data 2–4, respectively.

### Reporting summary
Further information on research design is available in the Nature Portfolio Reporting Summary linked to this article.

## Data availability
The cryo-EM density map generated in this study of the $_{dim}$APJR$^{AP13}$-Gi, $_{mon}$APJR$^{AP13}$-Gi, Apo APJR, JN241-APJR, JN241-9-APJR and JN241-9-APJR-Gi have been deposited in the Electron Microscopy Data Bank (EMDB) under accession code EMD-38574 (https://www.ebi.ac.uk/emdb/EMD-38574, $_{dim}$APJR$^{AP13}$-Gi), EMD-38575 (https://www.ebi.ac.uk/emdb/EMD-38575, $_{mon}$APJR$^{AP13}$-Gi), EMD-38578(https://www.ebi.ac.uk/emdb/EMD-38578, Apo APJR), EMD-38579 (https://www.ebi.ac.uk/emdb/EMD-38579, JN241-APJR), EMD-39810 (https://www.ebi.ac.uk/emdb/EMD-39810, JN241-9-APJR), EMD-39816 (https://www.ebi.ac.uk/emdb/EMD-39816, JN241-9-APJR-Gi), and model coordinates have been deposited in the Protein Data Bank (PDB) under accession number 8XQE (https://doi.org/10.2210/pdb8XQE/pdb, $_{dim}$APJR$^{AP13}$-Gi), 8XQF (https://doi.org/10.2210/pdb8XQF/pdb, $_{mon}$APJR$^{AP13}$-Gi), 8XQI (https://doi.org/10.2210/pdb8XQI/pdb, Apo APJR), 8XQJ (https://doi.org/10.2210/pdb8XQJ/pdb, JN241-APJR), 8Z74 (https://doi.org/10.2210/pdb8Z74/pdb, JN241-9-APJR) and 8Z7J (https://doi.org/10.2210/pdb8Z7J/pdb, JN241-9-APJR-Gi), respectively. All other data generated in this study are provided in the Supplementary Information, Supplementary Data 1–4 and Source Data files. Source data are provided with this paper.

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

## Acknowledgements

We thank Qiwen Tan, Qiaoyun Shi, Lu Zhang, Suwen Hu, Na Chen, Lin Wang, Fangfang Zhou, for protein cloning, expression and assay support; Li Wang, Dandan Liu, Qianqian Sun, Yuan Pei at the Bio-EM facility at ShanghaiTech University for data collection support. This work was supported by the National Natural Science Foundation of China (grants 32071194 to F.X.; 32330049 to J.L).

## Author contributions

Y.Y. and L.L. performed cloning and purification of APJR-Gi complexes with various ligands and APJR-sdAb complexes, performed cryo-EM sample preparation, data collection and structure analysis; L.W. performed cryo-EM data processing and structure determination; C.X. designed the single-molecule experiments, collected and analyzed the data and prepared the figure; S.L. and M.N. performed molecular dynamics analysis; Y.L. performed single-molecule photobleaching and T.Y. wrote the code for analysis with the guidance of W.J.; M.N. and M.Z. assisted in some protein purification work; Y.Y., M.N. and F.L. carried out Glo-Sensor cAMP assay; JL.L. assisted in protein expression work; S.S. and H.L. instructed the sdAb preparation; A.W., M.A.H. and R.C.S. instructed the manuscript preparation; F.X. conceived and supervised the project. J.L. participated in the interpretation of the data. Y.Y., L.L., C.X., J.L. and F.X. wrote the manuscript.

## Competing interests

S.S., H.L., M.A.H. and R.C.S. are current or former full-time employees and/or founders of Structure Therapeutics. The remaining authors declare no competing interests.
