## [Transparent Peer Review file · Nature Communications]

Structural insights into the regulation of monomeric and dimeric apelin receptor

Corresponding Author: Professor Fei Xu

Editorial Note: Parts of this Peer Review File have been redacted as indicated to remove third party material where no permission to publish were obtained.

Version 0:

Reviewer comments:

Reviewer #1

(Remarks to the Author)

The apelin-AJPR signaling is important for the regulation of cardiovascular function and fluid homeostasis. In previous studies, authors have shown the mixture of the 1:1 and 2:1 complex of the agonist-bound APJR-Gi and proposed the importance of dimerization of class A GPCR in the activation process. In the present work, they determined several APJR structures (apo-dimer, antagonist antibody-bound dimer, agonist-bound monomer, - dimer and agonist antibody bound-monomer) and proposed a model for the dimer to monomer conversion by Gi protein-coupled activation process of APJR. Overall, the dimer to monomer transition for the regulation of a class A GPCR proposed here is an interesting model that could be expand to other class A GPCRs. However, the data in the present work do not fully support their hypothesis. First, all the endogenous agonist-bound APJR structures (in previous and present studies) exhibited a mixture of monomer- and dimer APJR-Gi complex with the predominant populations of a dimer (even single molecule analysis result). There is no clear evidence that dimeric state is the only inactive form. More importantly, the present work suffers for the quality of the experimental maps and the reliability of the structures. In order to prove their model confidently, authors should provide better EM density maps and more reliable structures.

Questions

1. The major problem in this work is the reliability of the structures due to the low quality of the EM density maps. Although the stats of the global resolution are reasonable, good part is localized at the G protein part and maps at a number of regions are low quality. I carefully checked the maps and the most of the EM density maps authors provided have several invisible regions, yet authors modelled these regions (I do not know how authors did it). The maps for Ap13-bound monomer and Ap13-bound dimer (prot B) region are very poor; entire map suffers with low quality and residues that are critical for dimerization, and a number of other regions are not visible. For AP13 dimer, quality of the map near the ligand binding pocket are very poor – yet author claims this structure is partly activated – but with this quality of the map, one cannot distinguish with simple inactive apo form. In all the maps of apo, JN 241-9 (G-free and -bound) the map quality is very poor and should be significantly improved to confirm the conclusion authors have made. The antagonist-bound APJR map is the most reliable one among the provided models, but even this one exhibited inaccurate ICL.
2. The dimeric interface is critical for the activation of APJR. However, due to inaccuracy of the position of the dimerization motif (F97, F100, F101), it is unclear how structural transition induced by either agonist – and/or G-protein is transmitted to this part to induce the conformation change. In fact, the hydrophobic interface is very small and rotation of the side chains without influence from external forces could alter the ratio of monomer and dimer.
3. Also, how G-binding affects the dimeric interface is not clearly presented. With the correct model, authors should explain how G-protein binding alters the conformation in the dimeric interface and dissociates the dimeric APJR. For instance, in the fully active form – Jn241-9 Gi, AP13 Gi, 7W0P (F101 mutant), orientations of F100 and F101 differ from G-free forms. But no explanation was provided how such rearrangement can be made.
4. How similar (or different) in the G-protein binding to agonist antibody-APJR and F101-APJR mutant? In previous work in NSMB (2022), authors described the relative positions of G-protein with respect to different APJR states clearly. But in the present work, these are all missing.
5. In this reviewer's opinion, "the protB in Ap13-APJR dimer-Gi is a partially active state" is an over-interpretation. The authors claim that the H8 helix is disordered in this structure as one of the supporting evidences. However, most of the structures presented in this work showed invisible maps for the H8 helix. Furthermore, even ICL 2/3 are also disordered. Thus, it is likely that invisible H8 is due to low quality of the map (rather than intermediate structure).
6. Related with the above question, instead of interpreting this dimer as an intermediate structure from dimer to monomer

transition, one could simply interpret that one active monomer interacts with inactive apo monomer to form a dimer.

7. It is extremely unusual that the structure of apo-, agonist bound (G-free) and antagonist-bound APJR dimer structures are very similar (if not identical). Again, due to the inaccuracies in the ICLs and side-chains in TM regions, the model should be reconsidered. As the authors showed in the single molecule imaging in live cell analysis, a mixture of dimer and monomer APJR is present with the dominance of dimer. Only in artificial environments as shown here by some specific antibody, monomer is dominant – but this may not happen in the normal conditions.

8. The authors need to prove more convincingly the idea of negative cooperativity, and basis for the structural transmission from protomer A to protomer B, that makes the protein inactive.

9. line 173 to 175. Is it possible if the crystal structure indicates that the inactive antagonist-bound APJR also exists as a monomeric form? Thus, both monomer and dimer (at least) exist together as inactive form. How is the conformation (including dimeric interface) of APJR in the crystal structure?

10. line 217 ~ 224. "" Specifically, the binding of ligands such as apelin-13 to ProtAs may exert an inhibitory effect on the ligand binding and activation of ProtBs, while ligands like cmpd644 and ELA appear to have a lesser impact"" could be just a hypothesis.

I do not see sufficient structural and biochemical evidences to support authors model; The dimeric interface is very small (weak protomer A-B interaction) which would limit the regulation between the two protomers. How apelin-13 binding to Prot A exerts its effect on Prot B? Authors need to support this idea with more data or provide clear explanations.

11. line 253 - in the absence of Gi proteins, the agonistic antibody bound to APJR remains in an inactive state and predominantly exists in a dimeric state (Extended Data Fig. 8a). However, upon co expression of Gi proteins, a remarkable transition of APJR to an active state occurs, along with the significantly reduced dimeric species ""

Why G-free state is inactive? Do authors claim that APJR is inactive in the presence of agonist? If so, is there any example that the agonist-bound GPCR forms inactive state?

Minor concerns

1 line 173 Please cite the reference on previously reported JN241-bound APJR crystal structure.

2. What is the PDB ids that are used in Fig 4b (also please check others too).

3. line 205 what is the meaning of 'aligning with the ligand association on ProtB'? Do authors mean aligning Prot B (or aligning the ligand on Prot B; if the ligand alignment, how did they do ?.

4. Fig 1b and Extended Data Fig 8a. G-free protomer clashed with G-protein in the bound state of JN241-9. In the presence of apelin-13, a fraction of the G-bound GPCR dimer is retained in complex with G (Fig 1b). Are there any differences in the G-protein binding modes between the two states?

Reviewer #2

(Remarks to the Author)

The manuscript by Yue et al reports the cryo-EM structure of the apelin receptor (APJR) in complex with its main endogenous ligand, apelin-13 and the trimeric Gi protein. In addition, it also reports the cryo-EM structure of APJR in complex with a camelid nanobody (VHH) with antagonistic activity (JN241) on the receptor, and an agonist version of this nanobody (JN241-9) that stabilizes the complex between APJR and the trimeric Gi protein. It's worth noting that these JN241 and JN241-9 nanobodies have been described previously (Ma et al Science Advances 2020). And the crystal structure of the complex between APJR and JN241 was reported in this previous study. Four structures of APJR complexes are new: 1) apelin-13-APJR-Gi (stoichiometry 1:2:1); 2) JN241-9-APJR-Gi (stoichiometry 1:1:1); 3) JN241-9-APJR (stoichiometry 2:2) and 3) JN241-APJR (stoichiometry 2:2).

Major concerns:

1- The model proposed in Fig. 6 is highly hypothetical and based on the different new structures obtained in this study and validated only by a few molecular dynamics experiments.

2- Along the same line, the title is based on this highly hypothetical model described in Fig. 6.

3- It is intriguing to observe that the structure of the complex between APJR and apelin-13 shows mainly a dimer with a protomer that does not bind apelin-13, the one not coupled to the Gi protein (agonist:APJR:Gi stoichiometry of 1:2:1). This differs from similar agonist-APJR-Gi complexes obtained by the same group and first author (Yue et al NSMB 2022). Indeed, in the complexes between APJR and the other agonists, the small compound Cmp-644 and the endogenous peptide ELA-32, an APJR homodimer with each protomer liganded by an agonist was observed, with an agonist:APJR:Gi stoichiometry of 2:2:1. How can we explain the difference between this previous study and this new study with apelin-13?

4- cAMP and beta-arrestin dose-responses should be shown.

Reviewer #3

(Remarks to the Author)

This manuscript by Yue et al. reports new structures of the apelin receptor APJ bound to various ligands, including apelin-13, single-domain antibody fragments, and an apo structure. Several of these structures show the receptor in a dimeric state, including an asymmetric dimer with one receptor protomer bound to apelin-13 and G protein while the other is ligand-free. Structural data are complemented by some signaling assay results and some molecular dynamics simulations.

Overall the manuscript and the results presented are interesting and likely important given the biomedical relevance of the APJ signaling axis. The manuscript is a bit difficult to follow in places, with quite a large amount of fine structural detail described with limited discussion of broader context and importance. Shortening the manuscript for the sake of clearer focus may help improve the readability. Several important issues should be addressed, listed below:

Major points:

1. The cryoEM density for apelin-13 in Figure 1 appears rather poor. It is difficult to evaluate however given the very small images and low resolution.
2. Figure 1A contains a variety of vague statements, including "enhance muscle health" and "auto-aging". It is not clear what these really mean, and they should be removed or replaced with more precise descriptions. Terms like "vasodilation" have precise meanings and are more appropriate here. It is also worth noting that vasodilatory effects are not universally good or bad, but rather depend on context. Therapeutic activation of APJ may have toxicities, and the portrayal of APJ signaling as a panacea seems overly simplistic.
3. For data presented in Figure 2D was receptor expression level measured? This is an essential control to determine if differences in signaling reflect altered receptor activation, altered expression, or some combination of the two. The heat map is described as presenting mean \pm -SEM in the figure legend, but I don't understand how this is possible for the heat map. The meaning of the three columns is not described. The color code at the bottom has labels that are too small and low resolution to be readable.
4. The Trp6.48 flip shown in Fig. 3F should be supported by evidence from the cryoEM density, to verify that this flip is unambiguously clear.
5. In line 320-321 it is stated that this work has opened up new avenues for drug development. Is this really true? Have drugs been developed based on the work presented here? If not, a more measured statement would be more appropriate so as not to overstate the importance of the results.

Minor points:

1. The word "compelling" in line 40 is a subjective interpretation and probably should be omitted.
2. GSFSC plots are very small and low resolution, to the point of being partially unreadable

Reviewer #4

(Remarks to the Author)

In this manuscript, Yue et al. use cryo-EM to solve the structure of the apelin receptor (APJR) in both its apo and holo states (i.e. bound to its agonist, apelin, and to agonist/antagonistic antibodies), in the absence and presence of its cognate Gi protein. While the authors previously showed that APJR dimers coexist with monomers, in this study they further explored the dimerization mechanism of this receptor. Their results show that whereas the endogenous agonist of APJR only binds to the protomer engaged in G protein coupling, the antagonistic antibody do bind to both protomers, within a more compact dimer. On the other hand, the agonistic antibody seems to promote dimer formation in the absence of G protein. The authors hypothesize that G protein binding induces the dissociation of dimers into monomers. Their work is very interesting, and has a significant relevance in the GPCR field, and, specifically, regarding the dimerization mechanism of these receptors, which remains not well understood.

I have a major comment that, in my opinion, authors should address, and several minor ones that should hopefully help improve this work.

Major comments:

- Page 12 "Subsequently, upon binding to G proteins, the distance between the two protomers of APJR increases, supported by MD simulations indicating dimer destabilization upon agonist binding and G protein coupling"

MD simulations do not support an increase in the MD distance between protomers. The very short replicas (250 ns could easily be a good equilibration time for such complex system), only show that the apelin-13 bound system is not well equilibrated. In fact, one of the replicas show a clear decrease in distance (pink line) between protomers. I do not think one can state that MD sims support increase in distance observed in the cryo-EM structures. Authors should either perform proper MD simulations (i.e. independent, well-equilibrated and longer trajectories), or rather state that the results do not fully support the experimental findings.

Minor comments:

- Page 6: "Structural comparisons unveil similarities between the structures of ProtAAP13 and ProtAELA, with a root mean square deviation (RMSD) of 1.178 Å (Extended Data Fig. 4c)."

A better visual representation of the superimposition (i.e. more detailed cartoon representation) would allow a quick inspection of the results with more details.

- Page 9: "Interestingly, we observed a slight outward movement of the "toggle switch" W261 6.48 in ProtB, indicating a partial activation state compared to the inactive state (Extended Data Fig. 6e). Additionally, a minimal downward movement of the key residue Y2997.43 was noted, potentially linked to the partial activation of ProtB (Extended Data Fig. 6e). These structural changes resulted in an intermediate conformation of ProtBAP13, suggesting that ProtB may have a less favorable binding pocket for agonists compared to ProtA, which is stabilized by Gi- protein binding."

Do these results come from just visual inspection? Perhaps a thorough comparison of structures, including side chain movements would give other interesting details. This can be achieved using pretty basic computational tools, and would also help describing what authors describe as "intermediate".

- Page 9: "The asymmetric organization of Gi-protein coupling and ligand binding between the two protomers implies the involvement of an allosteric regulatory mechanism associated with the dimerization process"

I do not think that the results "imply" this mechanism, but rather suggest a potential allosteric regulation. I would tone it down.

- Page 9: "This suggests that ProtBs may act as allosteric modulators influencing downstream signaling pathways through ProtAs"

Unveiling the structural reason behind this modulation is asking for too much, but authors should speculate about the structural mechanism, based on a more advanced comparative analysis of the overall structure and/or the dimer interface.

How? I mean, is there any structural hint that could explain what is inducing the apo state of protomer B?

- Page 10: "Suggesting a destabilization of dimerization leading to a transition from dimer to monomer upon agonist binding and G-protein coupling."

- Page 11: "The conformational rotation of Y105 by approximately 45 degrees towards TM2/3 and TM6/7 in this context plays a crucial role in triggering receptor activation (Fig. 5e, f)."

Authors should state whether the simulations show this rotation, too, or, at least, this tendency.

- Page 11: "meticulous analysis of the agonistic antibody"

What do authors mean with meticulous? Could you be specific? What type of analysis?

- Methods (MD): "A missing loop in one receptor was reconstructed by borrowing the corresponding symmetrical loop from a homologous receptor"

What loop? What homologous receptor?

- Methods (MD):

The methodology for generating the replicas is not detailed in the methods. Are these replicas independent? Did authors build each replica independently in CHARMM-GUI? Otherwise, are they re-spawned from one system? This could be critical for the interpretation of results, specifically when one is interpreting the following 250 ns after a very short equilibration.

Version 1:

Reviewer comments:

Reviewer #1

(Remarks to the Author)

Overall, authors have addressed proper responses on most of this reviewer's concerns in the revised text and the quality of the cryo-EM maps has been improved compared to the previous one. Nevertheless, I do have some comments on the overstated sentences which are added during the revision.

Line 185-190. "These structural changes suggest a potential scenario.."

I am still not convinced that the binding of an inactive protomer to the active protomer negatively regulates the activity of APJR in response to the specific biological response. There is no clear evidence for such negative allosteric effect. It could be simply interaction between the active and inactive protomers over the physiological concentrations. In my opinion, authors should mention such a possibility or should tone down their proposal.

In discussion,

Line 315-326: ".....These findings indicate that the binding of the Gi proteins may impose allosteric effect to the conformations at the dimer interface, leading to a large-scale flexibility which in turn promotes the dissociation of the dimer..."

>> Some structures presented in this study are determined simply at low resolutions and it is difficult to reason that the low-resolution structures are related to the functional regulation of G-protein induced dissociation of the dimer. Authors have added this paragraph during the revision and no data support for this paragraph, which is clearly overstated.

Line 325-332: Authors presented their hypothetical working model (also in Ext Fig 9). This model could mislead the readers as if this is a general mechanism for the activation of APJR. In fact, only 6% increase of the APJR dimer has been observed in the presence of the small molecule agonist as shown in this manuscript. Moreover, the G-protein induced dimer to monomer conversion is observed only in specific agonist (JN241-9) and the APJR dimer-G complex is an active form in the presence of certain agonist. Thus, the authors should at least modify this paragraph to discuss the activation model in more conservative and careful manner.

-In addition, for the structures presented, the cryo-EM maps for ICLs are some ECLs are not visible, onto which authors built the models. I think authors should list possible disordered regions in the text. Moreover, the side-chains of the Q chain of the dimAPJ_AP13 are overfitted.

Reviewer #2

(Remarks to the Author)

The authors have addressed my concerns.

But I suggest clarifying the beginning of the title. Also, "APJR" should be replaced by "apelin receptor".

Reviewer #3

(Remarks to the Author)

The revised manuscript addresses some of the comments raised in the initial round of review, although the authors have largely chosen to do this through the removal of data rather than addressing technical comments with inclusion of additional or improved data. Specifically, the mutagenesis results were removed, rather than adding expression level controls. Do the mutants not actually express comparably? In the revised manuscript, essentially no functional data are presented, which makes it difficult to evaluate whether there is a meaningful biological insight here. The structural results are also somewhat poorly supported as was highlighted by other reviewers in the first round. Fig R5/Ex Data 7d show density for previously reported structures of APJ is poor, but this does little to increase confidence in the structures reported here. Fig R11/Ex Data 5g is similarly unclear, and likely compatible with other reasonable models of the Trp6.48 rotamer.

Overall, I believe the revised manuscript addresses some of the technical points raised by removing incompletely controlled data. The narrative is a bit clearer, although still somewhat difficult to follow (particularly in view of the lack of functional data). It is hard to conclude anything regarding biological importance of the results in the revised version of the manuscript.

Reviewer #4

(Remarks to the Author)

Thanks for addressing all points. I only have one comments regarding the major point of my revision

- The authors decided to not extend the simulations or perform new trajectories, but to state that simulations do not support the increase in distance between protomers. Please make sure this is clear across the text, for example, please remove "supported by MD simulations" from the following sentence of the revised manuscript (page 12, lines 310-312):

"Subsequently, upon binding to G proteins, the distance between the two protomers of APJR increases, supported by MD simulations indicating dimer destabilization"

Version 2:

Reviewer comments:

Reviewer #1

(Remarks to the Author)

Authors have properly addressed all concerns raised by this reviewer.

Reviewer #3

(Remarks to the Author)

The revised manuscript addresses some of the points raised. Data on expression levels are rather hidden, and should be emphasized more clearly in my view. I am still skeptical of the overall impact of this manuscript, although most of my major technical concerns are addressed. I hesitate about how important this insight is for a broad audience, especially in view of the numerous caveats.

We express our gratitude to the four reviewers for their diligent evaluation of our manuscript. Their constructive suggestions and comments have helped us a lot to improve the manuscript during the revision stage. Our point-by-point responses to each reviewer's comments are listed below in blue text. The textural changes in the revised manuscript are highlighted in the marked-up version. All the line numbers (in red) indicated in this rebuttal are referred to the ones in the marked-up version.

Reviewer's Comments:

Reviewer #1 (Remarks to the Author):

The apelin-APJR signaling is important for the regulation of cardiovascular function and fluid homeostasis. In previous studies, authors have shown the mixture of the 1:1 and 2:1 complex of the agonist-bound APJR-Gi and proposed the importance of dimerization of class A GPCR in the activation process. In the present work, they determined several APJR structures (apo-dimer, antagonist antibody-bound dimer, agonist-bound monomer, - dimer and agonist antibody bound- monomer) and proposed a model for the dimer to monomer conversion by Gi protein-coupled activation process of APJR.

Overall, the dimer to monomer transition for the regulation of a class A GPCR proposed here is an interesting model that could be expand to other class A GPCRs. However, the data in the present work do not fully support their hypothesis. First, all the endogenous agonist-bound APJR structures (in previous and present studies) exhibited a mixture of monomer- and dimer APJR-Gi complex with the predominant populations of a dimer (even single molecule analysis result). There is no clear evidence that dimeric state is the only inactive form. More importantly, the present work suffers for the quality of the experimental maps and the reliability of the structures. In order to prove their model confidently, authors should provide better EM density maps and more reliable structures.

Response: We are very grateful to the reviewer's overall positive evaluation and all the constructive comments. Following the valuable suggestions, we have done extensive revisions to address the major concerns on the map quality as structure reliability.

Questions

1. The major problem in this work is the reliability of the structures due to the low quality of the EM density maps. Although the stats of the global resolution are reasonable, good part is localized at the G protein part and maps at a number of regions are low quality. I carefully checked the maps and the most of the EM density maps authors provided have several invisible regions, yet authors modelled these regions (I do not know how authors did it). The maps for Ap13-bound monomer and Ap13-bound dimer (prot B) region are very poor; entire map suffers with low quality and residues that are critical for dimerization, and a number of other regions are not visible. For AP13 dimer, quality of the map near the ligand binding pocket are very poor – yet author claims this structure is partly activated – but with this quality of the map, one

cannot distinguish with simple inactive apo form. In all the maps of apo, JN 241-9 (G-free and -bound) the map quality is very poor and should be significantly improved to confirm the conclusion authors have made. The antagonist-bound APJR map is the most reliable one among the provided models, but even this one exhibited inaccurate ICL.

Response: We sincerely thank the reviewer for the constructive suggestion. Firstly, we have made further improvements to the cryo-EM maps and models of the apelin-13-bound-APJR-Gi complex structures. The cryo-EM map for ProtB in APJR dimers has been significantly enhanced, and following the reviewer's recommendation, we have interpreted ProtB as a simple inactive apo form. The related description is reflected in lines 185-190 and highlighted in our revised manuscript as follows:

lines 185-190:

These structural changes suggest a potential scenario where active ProtA interacts with inactive apo ProtB, leading to a reduced basal activity in regulation of the specific biological response. This is interpreted based on the “dimer-switch” mutagenesis and functional analysis results (PMID: 35817871 and Fig. R1) which demonstrated that the basal activity of APJR signaling is higher in its monomeric form compared to its dimeric form, in which ProtB might negatively modulate ProtA's basal activity.

Fig. R1 | Constitutive activity of WT-APJR and F101A mutant assessed by cAMP assay, from our previous work (PMID: 35817871); The F101A mutant showed higher basal activity (lower cAMP accumulation) than WT-APJR. [REDACTED]

To sharpen and concentrate our investigation into APJR dimerization, we condensed the content of our article by removing some structural details, particularly in regards to the apelin-13 binding mode. We have removed description of detailed interactions of apelin-13 as this has been comprehensively discussed in a recently published *Cell* paper by Zhang Y et al. (PMID: 38428423), and our results are highly consistent with their findings (Fig. R2). Only one notable difference is that the *Cell* paper didn't report the dimerization structure. We reasoned that one possible explanation might be concerning the modifications on the APJR's C-terminus (LgBit and double MBP tag) that may prevent the dimer to form.

Fig. R2 | Apelin-13 recognition mechanism from the recently published paper (PMID: 38428423); we observed the same interaction pattern in our structures. Therefore, detailed descriptions have been removed from the revised manuscript to avoid redundancy. [REDACTED]

Based on the reviewer's comments, we have made further improvements to the cryo-EM maps and models of the apo, JN241-9-APJR, and JN241-9-APJR-Gi complex structures to enhance their reliability.

For the JN241-9 bound structures, although we have further improved the overall resolution, the resolution of the antibody's extracellular domain remains relatively poor. However, our main focus is on the CDR3 region, which is inserted into the orthosteric pocket. We believe the cryo-EM density map for this region is reliable (**Fig. R3**).

Fig. R3 | Cryo-EM maps of CDR3 region of JN241-9 in the orthosteric pocket: (a) CDR3 region of JN241-9 in the Gi protein-bound structure; (b) CDR3 region of JN241-9 in the Gi protein-free structure.

For the cryo-EM map quality of ICLs in the antagonist-bound structure: due to the lack of Gi-protein support for the intracellular domain, the ICLs exhibit high flexibility in this structure, yielding relatively poor cryo-EM map at this region. However, since our study does not focus on the ICL domain, we believe that this inaccuracy is not significant.

2. The dimeric interface is critical for the activation of APJR. However, due to inaccuracy of the position of the dimerization motif (F97, F100, F101), it is unclear how structural transition induced by either agonist – and/or G-protein is transmitted to this part to induce the conformation change. In fact, the hydrophobic interface is very small and rotation of the side chains without influence from external forces could alter the ratio of monomer and dimer.

Response: We thank the reviewer for this suggestion. We have analyzed the cryo-EM densities of the dimer interface (FGTFF motif) from several dimer complex structures and presented them in **Fig. R4**, also shown in **Extended Data Fig. 3k**. We indeed observed that in the dimer complexed with G proteins, the EM map of the interface is not clearly defined; whereas in the absence of G-protein binding, the map at the dimer interface is more pronounced. This further supports the notion that the introduction of the G proteins may induce certain conformational changes at the dimer interface, leading to the increased dynamics of the dimer arrangement to facilitate the dissociation of dimer. We believe, as the reviewer has pointed out, that the hydrophobic dimerization interface, being very small, may be highly sensitive to subtle external variations. Therefore, we proposed that the combined action of the ligand and G proteins could lead to varying degrees of dimer dynamics and promote the dimer dissociation. We have included this in the Discussion from **lines 315 to 324** in the revised manuscript.

lines 315 to 324:

Additionally, we discovered that, due to the relatively low resolution, the accuracy of side-chain conformations at the dimer interface in the $\text{dimAPJR}^{\text{AP13}}\text{-Gi}$ complex as well as in the monomeric APJR-Gi (JN241-9-APJR-Gi and F101A-APJR^{ELA}-Gi) complexes is somewhat compromised. Conversely, all inactive forms without Gi protein binding (JN241-APJR, apo, and JN241-9-APJR) exhibited a clear cryo-EM map that confidently accommodates a compact dimer interface (Extended Data Fig. 3k and 7d). These findings indicate that the binding of the Gi proteins may impose allosteric effect to the conformations at the dimer interface, leading to a large-scale flexibility which in turn promotes the dissociation of the dimer.

Fig. R4 (also shown in updated Extended Data Fig. 3k) | Cryo-EM density maps and models of FGTFF motif (dimer interface) in $\text{dimAPJR}^{\text{AP13}}\text{-Gi}$, JN241-APJR, JN241-9-APJR and apo-APJR complexes, respectively.

3. Also, how G-binding affects the dimeric interface is not clearly presented. With the correct model, authors should explain how G-protein binding alters the conformation in the dimeric interface and dissociates the dimeric APJR. For instance, in the fully active form – Jn241-9 Gi, AP13 Gi, 7W0P (F101 mutant), orientations of F100 and F101 differ from G-free forms. But no explanation was provided how such rearrangement can be made.

Response: We appreciate the reviewer's constructive suggestions. We have analyzed the cryo-EM maps of the dimer interface (FGTFF motif) from dimer complex structures and presented them in **Fig. R4** and the FGTFF motif from the monomeric forms in **Fig. R5** (also shown in Extended Data Fig. 7d).

We discovered that, similar to the cryo-EM density map for the dimer interface in the $\text{dimAPJR}^{\text{AP13}}\text{-Gi}$ complex (**Fig. R4**), the active and monomeric APJR-Gi (JN241-9-APJR-Gi and F101A-APJR^{ELA}-Gi) complexes also exhibit somewhat compromised cryo-EM density map at the dimer interface (although these two complexes are monomers) (**Fig. R5**). We therefore discussed the potential conformational changes on the dimer interface between these structures and the important role of the binding of Gi proteins in promoting the dimer dissociation. This discussion can be found from **lines 315 to 324** in the revised manuscript (see our response to above **comment#2**).

Fig. R5 (also shown in updated Extended Data Fig. 7d) | Cryo-EM density maps and models of FGTFF motif (dimer interface) in JN241-9-APJR-Gi and F101A-APJR^{ELA}-Gi (PDB ID: 7W0P) complexes, respectively.

4. How similar (or different) in the G-protein binding to agonist antibody-APJR and F101-APJR mutant? In previous work in NSMB (2022), authors described the relative positions of G-protein with respect to different APJR states clearly. But in the present work, these are all missing.

Response: We appreciate the reviewer's careful evaluation and insightful comments. Following the reviewer's suggestions, we have compared the binding of G_i proteins between the JN241-9-APJR- G_i and F101A^{ELA}-APJR- G_i complexes. By analyzing the G_i proteins in the two complexes, we observed substantial rotations with the $G\alpha$ and $G\beta\gamma$ subunits, likely induced by the different ligands. The related analysis and illustrations have been supplemented in the revised manuscript (lines 237-243, highlighted in our revised manuscript as follows) and **Fig. R6** (also shown in Extended Data Fig. 7c).

Fig. R6 (also shown in updated Extended Data Fig. 7c) | Structural superposition of F101A-APJR^{ELA}- G_i (PDB ID: 7W0P) and JN241-9-APJR- G_i complexes. The conformational changes of $G\alpha$ and $G\beta\gamma$ were indicated as arrows.

lines 237-243:

We also performed the structural comparison between JN241-9-APJR- G_i complex and the monomeric mutant F101A^{ELA}-APJR- G_i structure that we have previously described¹ (PDB ID: 7W0P). The conformation of the receptor largely mirrors that observed in the prior structure. However, substantial rotations were observed with the $G\alpha$ and $G\beta\gamma$ subunits (Extended Data Fig. 7c). This rotation suggests that different ligands may induce distinct conformational changes in the G proteins, highlighting the potential for ligand-specific receptor activation profiles.

5. In this reviewer's opinion, "the protB in Ap13-APJR dimer- G_i is a partially active state" is an over-interpretation. The authors claim that the H8 helix is disordered in this structure as one of the supporting evidences. However, most of the structures presented in this work showed invisible maps for the H8 helix. Furthermore, even ICL 2/3 are also disordered. Thus, it is likely that invisible H8 is due to low quality of the map (rather than intermediate structure).

Response: We appreciate the reviewer's comment. We agreed with the reviewer that we may have over-interpreted the conformational state of ProtB. Accordingly, we have deleted the corresponding description in the revised manuscript. In line with our

response to above comment#1, we simply defined the conformational state of ProtB as an inactive apo form.

6. Related with the above question, instead of interpreting this dimer as an intermediate structure from dimer to monomer transition, one could simply interpret that one active monomer interacts with inactive apo monomer to form a dimer.

Response: We thank the reviewer for this suggestion. We have revised this part and interpret as that one active monomer interacts with inactive apo monomer to form a dimer, lines 185-190 in the revised manuscript (see our response to above comment#1).

7. It is extremely unusual that the structure of apo-, agonist bound (G-free) and antagonist-bound APJR dimer structures are very similar (if not identical). Again, due to the inaccuracies in the ICLs and side-chains in TM regions, the model should be reconsidered. As the authors showed in the single molecule imaging in live cell analysis, a mixture of dimer and monomer APJR is present with the dominance of dimer. Only in artificial environments as shown here by some specific antibody, monomer is dominant – but this may not happen in the normal conditions.

Response: We thank the reviewer for this comment. Firstly, it is important to acknowledge the common occurrence where the apo (unbound) forms of GPCRs share structural similarities with their ligand-bound states due in part to the absence of binding to G proteins or other partner molecules. A similar phenomenon has been observed and reported before. For example, in the case of orphan GPR52, the two states: apo (PDB ID: 6LI1) and ligand-bound (PDB ID: 6LI0), when both in the absence of G proteins, are highly similar.

Moreover, in addressing the observation of antibody-induced monomer dominance, one possible interpretation is that given the higher stability and signaling activity of monomer than dimer, an agonistic antibody with longer residence time than peptides or small molecules, may preferably stabilize the Gi-coupled APJR in the monomer (more stable) form. Meanwhile, we acknowledge the notion that we have observed the dominance of dimer in the over-expression cell systems. Nevertheless, the agonistic antibody induced monomer dominance is reasonable at such over-expression system, as it represents the most stable and signaling-active form when coupled to G proteins.

8. The authors need to prove more convincingly the idea of negative cooperativity, and basis for the structural transmission from protomer A to protomer B, that makes the protein inactive.

Response: We thank the reviewer for the comment. We discussed the potential structural transmission mechanism between ProtA and ProtB from the perspective of G protein binding and ligand property, details of which are provided in lines 268-283 in the Discussion section, also as follows:

lines 268-283:

We inferred that in the absence of Gi-protein binding, the agonist may interact with

both orthosteric sites within the dimer. Upon G_i protein coupling, ProtA, now complexed with the G_i protein, effectively stabilizes the ligand at its binding site. This aligns with the mechanism through which G-proteins mediate the enhancement of agonist activity [PMID: 27362234; 4726]. In contrast, ProtB, without G-protein interaction, potentially suffers from reduced ligand stability in its binding pocket, potentially leading to ligand release and rendering ProtB inactive. For ligands with limited binding affinities with APJR, or those as flexible as apelin-13, ProtAs might exert an inhibitory effect on the ligand binding and activation of ProtBs. Conversely, ligands with higher affinity (such as cmpd644, PDB ID: 7W0L, PMID: 35817871), or those with extensive interactions, (such as ELA, PDB ID: 7W0N, PMID: 35817871), seem to have minimal impact as they can still bind to ProtBs. While this hypothesis requires further exploration, it offers a plausible mechanism explaining how G-protein binding enhances agonist binding to the receptors (ProtAs) whereas ProtB may function as a negative allosteric modulator, reducing the overall signaling output when forming the dimer with ProtA.

9. line 173 to 175. Is it possible if the crystal structure indicates that the inactive antagonist-bound APJR also exists as a monomeric form? Thus, both monomer and dimer (at least) exist together as inactive form. How is the conformation (including dimeric interface) of APJR in the crystal structure?

Response: We thank the reviewer for raising the question. The JN241-bound APJR crystal structure exists in a monomeric form, which may be owing to the two facts: 1) the APJR construct used for the crystallization suffers from extensive mutations and engineering which may introduce artifacts to the conformation and oligomeric states; 2) the crystallization conditions and crystal packings may alter the oligomeric states of the APJR. However, we cannot exclude the possibility that monomers and dimers may coexist as inactive forms. A comparison of our dimeric structure with the crystal structure reveals overall similarity in the receptor portions. Additionally, we have analyzed the dimer interface and addressed our findings in the text (lines 151-158, highlighted in our revised manuscript as follows), as shown in **Fig. R7a, b** (also shown in Extended Data Fig. 5d, e).

lines 151-158:

Upon comparing our cryo-EM structure with the crystal structure, we observed that APJR adopts a similarly inactive conformation (Extended Data Fig. 5d). Delving further into the dimer interface, we noted that the interface map density accommodates the five amino acids of the FG₂TFF motif with high clarity in the cryo-EM map (Extended Data Fig. 5e). Compared to the FG₂TFF motif in the crystal structure, we detected subtle conformational shifts in the side chains of three pivotal phenylalanine residues. These slight changes are likely due to dimerization effects in the cryo-EM structures.

Fig. R7 (also shown in updated Extended Data Fig. 5d, e) | Structural comparison between JN241-APJR cryo-EM and crystal (PDB ID: 6KNM) structures. **a**, Overall structural comparison between JN241-APJR cryo-EM and crystal structures. **b**, Superimposition of the crystal structure of JN241-APJR onto cryo-EM structure, with a focus on the dimeric interface of FGTF motif, shows subtle conformational changes in the side chains of three critical phenylalanine residues. Additionally, the cryo-EM density map for the amino acids comprising the FGTF motif is shown.

10. line 217 ~ 224. “” Specifically, the binding of ligands such as apelin-13 to ProtAs may exert an inhibitory effect on the ligand binding and activation of ProtBs, while ligands like cmpd644 and ELA appear to have a lesser impact”” could be just a hypothesis.

I do not see sufficient structural and biochemical evidences to support authors model; The dimeric interface is very small (weak protomer A-B interaction) which would limit the regulation between the two protomers. How apelin-13 binding to Prot A exerts its effect on Prot B? Authors need to support this idea with more data or provide clear explanations.

Response: We thank the reviewer for the comment. A notable feature of the dimeric APJR structure is its small dimer interface, which, nevertheless, comprises three pairs of Π - Π interactions, suggesting a reasonably robust interaction. We hypothesize that the modulation of this interaction could be mediated through the binding of G proteins, as elaborated above in response to comment #8.

11. line 253 – “in the absence of G_i proteins, the agonistic antibody bound to APJR remains in an inactive state and predominantly exists in a dimeric state (Extended Data Fig. 8a). However, upon co expression of G_i proteins, a remarkable transition of APJR to an active state occurs, along with the significantly reduced dimeric species”
Why G-free state is inactive? Do authors claim that APJR is inactive in the presence of agonist? If so, is there any example that the agonist-bound GPCR forms inactive state?

Response: We thank the reviewer for the comment. In the absence of G proteins, receptors may remain effectively unresponsive despite being bound to agonists, thus retaining an inactive state. Some GPCRs may be readily activated, requiring only the

presence of agonists to achieve a partially or fully active conformation—for instance, the CB1 receptor or the A2a receptor (PMID: 28678776, PMID: 21393508), as observed in resolved crystal structures. However, the activation of some GPCRs may necessitate a substantial energy threshold, implying that agonists alone may not suffice to induce a partial or full activation; the binding of G proteins could be paramount in achieving an activated state. For example, the orphan GPR52 exhibits inactive-like conformation in the presence or absence of agonist, when the G-protein is not coupled (PMID: 32076264), as shown in the two crystal structures (apo: PDB ID 6LI1; agonist-bound: PDB ID 6LI0); while exhibiting active conformation upon G-protein binding even in the absence of an agonist (PDB ID: 6LI3). Similar observations for the 5HT2B receptor: when bound to the agonist LSD and lacking a transducer, the receptor is positioned in an intermediate state with less pronounced displacement of TM6 (PMID: 36087581). Additionally, the calcitonin gene-related peptide (CGRP) receptor, a class B1 GPCR, exhibits similar characteristics when bound with its neuropeptide ligand (PMID: 33602864). Further, our search on GPCRdb revealed that there are other GPCRs that remain in an inactive state after binding with agonists in the absence of G protein interaction (**Fig. R8**). It is worth considering, however, that some X-ray crystal structures might artificially present receptors in an inactive state due to the introduction of certain mutations.

receptor family	CL	Species	Method	PDB	Refined structure	Resolution	Preferred chain	State	Degree active (%)	% of Seq	Name	Type	Function
MRGR2	MARGPRD	Human	Cryo-EM	7Y14	7Y14_refined	3.2	R	Intermediate	55	82	β -alanine	small-molecule	Agonist
ACM1	M1	Human	X-ray	6ZF2	6ZF2_refined	2.2	A	Inactive	2	62	CHEMBL3354085	small-molecule	Agonist
GRM1	mGlu1	Human	Cryo-EM	7DGE	7DGE_refined	3.7	A	Intermediate	-	66	quisqualate	small-molecule	Agonist
GRM4	mGlu4	Human	Cryo-EM	7E9H	-	4.0	R	Intermediate	-	84	L-serine-O-phosphate	small-molecule	Agonist
CALRL	calcitonin-like	Human	Cryo-EM	7KNU	7KNU_refined	3.5	R	Inactive	10	74	Calcitonin gene-relate...	peptide	Agonist
NTR1	NTS1	Rat	X-ray	6Z8N	6Z8N_refined	2.8	A	Intermediate	15	80	(2-[S]-4-methyl-2-[(1...	small-molecule	Agonist
GABR2	GABA _{B2}	Human	Cryo-EM	6UO9	-	4.8	B	Intermediate	-	73	CHEMBL112710	small-molecule	Agonist
MTR1A	MT1	Human	X-ray	6ME2	6ME2_refined	2.8	A	Inactive	1	83	ramelteon	small-molecule	Agonist
MTR1B	MT2	Human	X-ray	6ME7	6ME7_refined	3.2	A	Inactive	1	79	CHEMBL15000	small-molecule	Agonist
GRM5	mGlu5	Human	Cryo-EM	6N51	6N51_refined	4.0	B	Intermediate	-	65	quisqualate	small-molecule	Agonist
PTH1R	PTH1	Human	X-ray	6FJ3	-	2.5	A	Inactive	0	62	ePTH	peptide	Agonist (partial)
GLR	glucagon	Human	X-ray	5YQZ	5YQZ_refined	3.0	R	Inactive	2	83	NNC1702	peptide	Agonist (partial)
GLP1R	GLP-1	Human	X-ray	5NX2	-	3.7	A	Intermediate	29	84	Truncated protein ago...	peptide	Agonist
APJ	apelin	Human	X-ray	5VBL	-	2.6	B	Inactive	4	79	AMG3054	peptide	Agonist
5HT2B	5-HT _{2B}	Human	X-ray	5TVN	5TVN_refined	2.9	A	Intermediate	44	61	Lysergide	small-molecule	Agonist
NTR1	NTS1	Rat	X-ray	4BUO	4BUO_refined	2.8	A	Inactive	3	72	neurotensin	peptide	Agonist

Fig. R8 | GPCRs that remain in an inactive state after binding with agonists in the absence of G protein interaction revealed by GPCRdb.

In the case of APJR, current structural analyses suggest that the receptor is unlikely to reach an active state with agonist binding alone, as indicated by the conformations of TM5 and TM6, which exhibit the typical inactive-like conformation with the lack of outward movement of TM6, thus implying that the presence of G proteins is crucial for

the activation of APJR. This phenomenon has already been observed in our first reported APJR structure (Ma, Yue et al, Structure, 2017, PMID: 28528775), where an apelin-mimic peptide (a potent agonist)-bound APJR crystal structure is stabilized in an inactive conformation. Accordingly, we have also supplemented our manuscript with a discussion on the importance of G proteins in the activation of the APJR (lines 255-263, highlighted in our revised manuscript as follows).

lines 255-263:

The activation of GPCRs is commonly acknowledged to necessitate the engagement of agonists and G proteins. While certain GPCRs can achieve full or partial activation through the sole action of agonists (PMID: 28678776; 21393508), others necessitate G protein binding to transition into an active state (PMID: 36087581; 33602864). Our structural analyses indicate that, in the case of APJR, agonist binding alone may not be capable of achieving the full activation as evidenced by the conformations of the receptors we have observed before (PMID: 28528775) as well as in this study: the TM6 lacks outward movement typically seen in class-A GPCR activation (Extended Data Fig. 7a). These findings underline the critical role of G proteins in facilitating APJR activation.

Minor concerns

1 line 173 Please cite the reference on previously reported JN241-bound APJR crystal structure.

Response: We thank the reviewer for this suggestion. We have cited the reference (PMID: 35817871) as requested in **line 151**.

2. What is the PDB ids that are used in Fig 4b (also please check others too).

Response: We thank the reviewer for this suggestion. Since we have removed the description related to Fig. 4b, we have also deleted Fig. 4b accordingly. Additionally, we have reviewed and addressed similar issues throughout the manuscript.

3. line 205 what is the meaning of 'aligning with the ligand association on ProtB'? Do authors mean aligning Prot B (or aligning the ligand on Prot B; if the ligand alignment, how did they do ?.

Response: We thank the reviewer for this suggestion. In the original manuscript, we intended to align the ProtB, not the ligand on ProtB. However, in order to more accurately describe the content of our article and **in response to comment #5** raised by the reviewer, we have deleted this sentence in the corresponding sections.

4. Fig 1b and Extended Data Fig 8a. G-free protomer clashed with G-protein in the bound state of JN241-9. In the presence of apelin-13, a fraction of the G-bound GPCR dimer is retained in complex with G (Fig 1b). Are there any differences in the G-protein binding modes between the two states?

Response: We apologized for the confusing description. We conducted a

comprehensive comparative analysis of the complex structures between JN241-9-APJR-Gi and $\text{dimAPJR}^{\text{AP13}}\text{-Gi}$. By aligning the receptors that bind to G proteins, we noticed that in JN241-9-APJR-Gi, both the $\text{G}\alpha$ and $\text{G}\beta\gamma$ subunits exhibited upward shifts. Such changes might conflict with ProtB in $\text{dimAPJR}^{\text{AP13}}\text{-Gi}$. The corresponding content (lines 233-237, highlighted in our revised manuscript as follows) and figure (Fig. R9, also shown in updated Extended Data Fig. 7b) are displayed in the revised manuscript.

lines 233-237:

By comparing the G proteins in JN241-9-APJR-Gi and $\text{dimAPJR}^{\text{AP13}}\text{-Gi}$ complexes, we observed that, in the JN241-9-APJR-Gi structure, the absence of ProtB results in a substantial relocation of the $\text{G}\alpha$ and $\text{G}\beta\gamma$ subunits towards the region that would otherwise be occupied by ProtB in the $\text{dimAPJR}^{\text{AP13}}\text{-Gi}$ complex (Extended Data Fig. 7b).

Fig. R9 (also shown in updated Extended Data Fig. 7b) | Structural superposition of $\text{dimAPJR}^{\text{AP13}}\text{-Gi}$ and JN241-9-APJR-Gi complexes shows conformational shifts in $\text{G}\alpha$ and $\text{G}\beta\gamma$, indicated by arrows. These shifts suggest an upward displacement in the JN241-9-APJR-Gi complex towards the region that would otherwise be occupied by ProtB in the $\text{dimAPJR}^{\text{AP13}}\text{-Gi}$ complex.

Reviewer #2 (Remarks to the Author):

The manuscript by Yue et al reports the cryo-EM structure of the apelin receptor (APJR) in complex with its main endogenous ligand, apelin-13 and the trimeric Gi protein. In addition, it also reports the cryo-EM structure of APJR in complex with a camelid nanobody (VHH) with antagonistic activity (JN241) on the receptor, and an agonist version of this nanobody (JN241-9) that stabilizes the complex between APJR and the trimeric Gi protein. It's worth noting that these JN241 and JN241-9 nanobodies have been described previously (Ma et al Science Advances 2020). And the crystal structure of the complex between APJR and JN241 was reported in this previous study. Four structures of APJR complexes are new: 1) apelin-13-APJR-Gi (stoichiometry 1:2:1); 2) JN241-9-APJR-Gi (stoichiometry 1:1:1); 3) JN241-9-APJR (stoichiometry 2:2) and 3) JN241-APJR (stoichiometry 2:2).

Major concerns:

1- The model proposed in Fig. 6 is highly hypothetical and based on the different new structures obtained in this study and validated only by a few molecular dynamics experiments.

Response: We appreciate the reviewer's comment. Currently, our model is developed primarily from a limited set of existing structural data and molecular dynamics (MD) simulations. While this model sheds light on the role of G proteins in modulating the dimerization and activation of APJR, it is acknowledged that additional experimental verification is likely required to fully substantiate its accuracy and applicability. Consequently, we have moved this model to the extended figures and made certain modifications and adjustments to the manuscript within the discussion part (in lines 325-332 and 335-336 and also shown as below).

lines 325-332, 335-336:

Consequently, based on the landscape of structures reported here and previously¹, along with MD simulation analysis, we propose a hypothetical working model of dimerization-modulated activation pathway for APJR. This model spans from the ligand-free state of APJR (symmetric dimer) to the antagonist-bound state (symmetric dimer), then to the agonist-bound state without G protein coupling (symmetric dimer dominance), and finally to the fully activated state with both agonist binding and G protein interaction (coexistence of asymmetric dimer and monomer, with monomer dominance in the case of an agonistic antibody-bound state) (Extended Data Fig. 9).....

Nevertheless, the model merits further research to elucidate and validate these processes.

2- Along the same line, the title is based on this highly hypothetical model described in Fig. 6.

Response: We thank the reviewer for the comment. We have made revision to the

title of the manuscript as follows:

Title: The Versatile Regulation of APJR Mediated by Dimerization, Ligand Binding, and G-Protein Coupling

3- It is intriguing to observe that the structure of the complex between APJR and apelin-13 shows mainly a dimer with a protomer that does not bind apelin-13, the one not coupled to the Gi protein (agonist:APJR:Gi stoichiometry of 1:2:1). This differs from similar agonist-APJR-Gi complexes obtained by the same group and first author (Yue et al NSMB 2022). Indeed, in the complexes between APJR and the other agonists, the small compound Cmp-644 and the endogenous peptide ELA-32, an APJR homodimer with each protomer ligand by an agonist was observed, with an agonist:APJR:Gi stoichiometry of 2:2:1. How can we explain the difference between this previous study and this new study with apelin-13?

Response: We thank the reviewer for the comment. Firstly, with the current three structures available, it is not yet possible to provide a definitive structural explanation regarding the ligand occupancy on the ProtBs. We reasoned that the binding of the agonist to the ProtB may be subject to greater flexibility than the one on ProtA due to the reduced binding stability in the absence of direct G-protein interaction. Additionally, since cmpd644 exhibits exceptionally high potency and ELA-32 engages extensive interactions with APJR, we hypothesize that this may contribute to the relatively higher stability of the ligand to bind to ProtBs.

Regarding ELA-32 and apelin-13, they exhibit similar potency on the G protein signaling. The ELA-32 comprises 32 amino acids and contains a disulfide bond, likely making it more stable overall compared to apelin-13. Moreover, the binding modes of ELA-32 and apelin-13 differ slightly. Taken together, these factors may contribute to the reduced stability of apelin-13 in the ProtB pocket due to its lack of G protein binding.

4- cAMP and beta-arrestin dose-responses should be shown.

Response: We thank the reviewer for the comment. To sharpen and concentrate our investigation into APJR dimerization, we have decided to omit certain detailed interactions of apelin-13, as our findings are highly consistent to those elucidated in a recently published Cell paper (PMID: 38428423) (Fig. R1, in response to Reviewer#1's comment#1). Therefore, all the original functional data to validate apelin-13 binding mode (cAMP and beta-arrestin assays) and related content have been deleted in the revised manuscript.

Our revised manuscript solely focuses on underling the APJR dimerization and regulation of the signaling pathways. It is worth noting that our finding contrasts with the above-mentioned apelin-13 paper (PMID: 38428423), where construct modifications at the C-terminus of the APJR (MBP and LgBit fusions) might have led to the absence of the dimerization phenomenon. This is just our speculation, though, as the construct we employed is the WT APJR sequence (without C-term fusion).

Reviewer #3 (Remarks to the Author):

This manuscript by Yue et al. reports new structures of the apelin receptor APJ bound to various ligands, including apelin-13, single-domain antibody fragments, and an apo structure. Several of these structures show the receptor in a dimeric state, including an asymmetric dimer with one receptor protomer bound to apelin-13 and G protein while the other is ligand-free. Structural data are complemented by some signaling assay results and some molecular dynamics simulations.

Overall the manuscript and the results presented are interesting and likely important given the biomedical relevance of the APJ signaling axis. The manuscript is a bit difficult to follow in places, with quite a large amount of fine structural detail described with limited discussion of broader context and importance. Shortening the manuscript for the sake of clearer focus may help improve the readability. Several important issues should be addressed, listed below:

Response: We appreciate the reviewer's positive evaluation and constructive suggestions. To sharpen and concentrate our investigation into APJR dimerization, we condensed the content of our article by removing some structural details, particularly in regards to the apelin-13 binding mode. As this has been comprehensively discussed in a recently published Cell paper (PMID: 38428423), and our results are highly consistent with their findings. Only one notable difference is that they didn't report the dimerization structure. We reasoned that one possible explanation might be concerning the modifications on the APJR's C-terminus in that paper (LgBit and double MBP fusion) that may prevent the dimer to form.

Meanwhile, we broadened our **analysis of the interactions between G proteins and dimers**. We have included the cryo-EM density map of the dimer interface and concentrated on APJR dimerization structural analysis with different ligand-bound states. Additionally, we have expanded our discussion on how **G protein influences the dimerization and activation of APJR**.

Following the reviewer's comments and suggestions, we have made extensive revisions throughout the manuscript, detailed below:

Major points:

1. The cryoEM density for apelin-13 in Figure 1 appears rather poor. It is difficult to evaluate however given the very small images and low resolution.

Response: We thank the reviewer for the comment. We have streamlined our analysis on the apelin-13 binding site, resulting in the removal of detailed descriptions regarding this aspect. Accordingly, we have also omitted the cryo-EM density map of apelin-13 from our study. This decision was made to sharpen the focus of our research towards more impactful findings—unlocking the scientific puzzle of APJR dimerization and signaling regulation, and to ensure that our study remains concise and highly relevant to the main focus.

Meanwhile, for other structure illustrations, we have re-made figures to provide clearer visualization, which is shown in **Fig. 1b** in the revised manuscript.

2. Figure 1A contains a variety of vague statements, including “enhance muscle health” and “anti-aging”. It is not clear what these really mean, and they should be removed or replaced with more precise descriptions. Terms like “vasodilation” have precise meanings and are more appropriate here. It is also worth noting that vasodilatory effects are not universally good or bad, but rather depend on context. Therapeutic activation of APJ may have toxicities, and the portrayal of APJ signaling as a panacea seems overly simplistic.

Response: We thank the reviewer for the comment. We have made changes to the Fig. 1a in the revised manuscript, also shown below as in Fig. R10. The therapeutic applications of APJR activation were summarized from previous literatures (PMID: 10617103; 23943882; 37833484; 11336787; 15907343). The figure was created with BioRender.com.

Fig. R10 (also shown in updated Fig. 1a) | A model demonstrating the potential pharmacological effects mediated by APJR activation.

3. For data presented in Figure 2D was receptor expression level measured? This is an essential control to determine if differences in signaling reflect altered receptor activation, altered expression, or some combination of the two. The heat map is described as presenting mean \pm SEM in the figure legend, but I don’t understand how this is possible for the heat map. The meaning of the three columns is not described. The color code at the bottom has labels that are too small and low resolution to be readable.

Response: We thank the reviewer for the comment. In an effort to refine the focus of our manuscript, and due to the relatively poor cryo-EM density map of apelin-13 despite of efforts on improving the map resolution, we have reduced our analysis of the apelin-13 binding pocket. Our attention has been directed towards a comparative study of ligand-induced dimerization. These reductions in the scope of our investigation were necessary to maintain a sharp narrative and to ensure that our findings remain focused on contributing substantial advancements to the understanding of APJR dimerization and signaling regulation. As a result, the original Figure 2D and related experimental

data have been removed from the revised manuscript.

4. The Trp6.48 flip shown in Fig. 3F should be supported by evidence from the cryoEM density, to verify that this flip is unambiguously clear.

Response: We thank the reviewer for the constructive comment. We have supplemented the cryo-EM density map of the amino acid W261^{6.48} in **Fig. R11** (also shown in updated **Extended Data Fig. 5g**) to provide a clearer visualization.

Fig. R11 (also shown in updated **Extended Data Fig. 5g**) | Cryo-EM density maps for the "toggle switch" residue W261^{6.48} within the JN241-APJR complex and dimAPJR^{AP13}-Gi complex structures, respectively.

5. In line 320-321 it is stated that this work has opened up new avenues for drug development. Is this really true? Have drugs been developed based on the work presented here? If not, a more measured statement would be more appropriate so as not to overstate the importance of the results.

Response: We thank the reviewer for the comment. As our focus is directed toward the regulation of dimerization and the impact of G proteins on dimers, we have accordingly removed the statement regarding the guidance for drug development.

Minor points:

1. The word "compelling" in line 40 is a subjective interpretation and probably should be omitted.

Response: We thank the reviewer for the comment. We have omitted the word "compelling".

2. GSFSC plots are very small and low resolution, to the point of being partially unreadable

Response: We thank the reviewer for the comment. We have increased the resolution of the GSFSC curve graph, see **Extended Data Figs. 2 and 4**.

Reviewer #4 (Remarks to the Author):

In this manuscript, Yue et al. use cryo-EM to solve the structure of the apelin receptor (APJR) in both its apo and holo states (i.e. bound to its agonist, apelin, and to agonistic/antagonistic antibodies), in the absence and presence of its cognate Gi protein. While the authors previously showed that APJR dimers coexist with monomers, in this study they further explored the dimerization mechanism of this receptor. Their results show that whereas the endogenous agonist of APJR only binds to the protomer engaged in G protein coupling, the antagonistic antibody do bind to both protomers, within a more compact dimer. On the other hand, the agonistic antibody seems to promote dimer formation in the absence of G protein. The authors hypothesize that G protein binding induces the dissociation of dimers into monomers.

Their work is very interesting, and has a significant relevance in the GPCR field, and, specifically, regarding the dimerization mechanism of these receptors, which remains not well understood.

I have a major comment that, in my opinion, authors should address, and several minor ones that should hopefully help improve this work.

Major comments:

- Page 12 "Subsequently, upon binding to G proteins, the distance between the two protomers of APJR increases, supported by MD simulations indicating dimer destabilization upon agonist binding and G protein coupling"

MD simulations do not support an increase in the distance between protomers. The very short replicas (250 ns could easily be a good equilibration time for such complex system), only show that the apelin-13 bound system is not well equilibrated. In fact, one of the replicas show a clear decrease in distance (pink line) between protomers. I do not think one can state that MD sims support increase in distance observed in the cryo-EM structures. Authors should either perform proper MD simulations (i.e. independent, well-equilibrated and longer trajectories), or rather state that the results do not fully support the experimental findings.

Response: We thank the reviewer for the constructive comment. Due to constraints on time and computational resources, we did not repeat the MD simulations. Following the reviewer's suggestion, we have supplemented with an explanation that the MD results do not fully support our experimental findings in the revised manuscript in **lines 219-220**, as highlighted below:

lines 219-220:

However, these MD simulation results could not conclusively support the role of G-protein in facilitating the dissociation of the APJR dimer. Thus, to further validate this hypothesis, we solved the cryo-EM structures of the agonistic antibody JN241-9 bound APJR in the presence or absence of Gi proteins and conducted comprehensive structural analysis (Fig. 5a, b and Extended Data Fig. 4e-l).

Minor comments:

- Page 6: "Structural comparisons unveil similarities between the structures of ProtAAP13 and ProtAELA, with a root mean square deviation (RMSD) of 1.178 Å (Extended Data Fig. 4c)."

A better visual representation of the superimposition (i.e. more detailed cartoon representation) would allow a quick inspection of the results with more details.

Response: We thank the reviewer for the comment. For better visualization and inspection, we have incorporated structural comparisons from the extracellular and intracellular views between ProtA^{AP13} and ProtA^{ELA}. See **Fig. R12**, also shown in the updated **Fig. 1c-e** in the revised manuscript.

Fig. R12 (also shown in updated Fig. 1c-e) | Structural comparison between dimAPJR^{AP13}-Gi and dimAPJR^{ELA}-Gi (PDB ID: 7W0N) in overall side view (a), intracellular view (b), and extracellular view (c), respectively.

- Page 9: "Interestingly, we observed a slight outward movement of the "toggle switch" W261 6.48 in ProtB, indicating a partial activation state compared to the inactive state (Extended Data Fig. 6e). Additionally, a minimal downward movement of the key residue Y2997.43 was noted, potentially linked to the partial activation of ProtB (Extended Data Fig. 6e). These structural changes resulted in an intermediate conformation of ProtBAP13, suggesting that ProtB may have a less favorable binding pocket for agonists compared to ProtA, which is stabilized by Gi- protein binding."

Do these results come from just visual inspection? Perhaps a thorough comparison of structures, including side chain movements would give other interesting details. This can be achieved using pretty basic computational tools, and would also help describing what authors describe as "intermediate".

Response: We thank the reviewer for the comment. We have revised the section of the paper, by following Reviewer#1's suggestions, we have described the ProtB state as "inactive apo monomer". Accordingly, we have supplemented the content with additional details to support this characterization and highlighted in the revised manuscript in **lines 185-190**, as follows:

lines 185-190:

These structural changes suggest a potential scenario where active ProtA interacts with

inactive apo ProtB, leading to a reduced basal activity in regulation of the specific biological response. This is interpreted based on the “dimer-switch” mutagenesis and functional analysis results (PMID: 35817871 and Fig. R1) which demonstrated that the basal activity of APJR signaling is higher in its monomeric form compared to its dimeric form, in which ProtB might negatively modulate ProtA’s basal activity.

- Page 9: "The asymmetric organization of Gi-protein coupling and ligand binding between the two protomers implies the involvement of an allosteric regulatory mechanism associated with the dimerization process"

I do not think that the results “imply” this mechanism, but rather suggest a potential allosteric regulation. I would tone it down.

Response: We thank the reviewer for the comment. We have changed to “...between the two protomers suggest potential involvement of ...” (see line 192 in the revised manuscript).

- Page 9: "This suggests that ProtBs may act as allosteric modulators influencing downstream signaling pathways through ProtAs"

Unveiling the structural reason behind this modulation is asking for too much, but authors should speculate about the structural mechanism, based on a more advanced comparative analysis of the overall structure and/or the dimer interface.

How? I mean, is there any structural hint that could explain what is inducing the apo state of protomer B?

Response: We thank the reviewer for the constructive comment. We hypothesize that the modulation of this interaction could be mediated through the binding of G proteins, as mentioned above. Further explanation on this aspect is provided in the revised manuscript in lines 268-283, highlighted in our revised manuscript as follows:

lines 268-283:

We inferred that in the absence of Gi-protein binding, the agonist may interact with both orthosteric sites within the dimer. Upon Gi protein coupling, ProtA, now complexed with the Gi protein, effectively stabilizes the ligand at its binding site. This aligns with the mechanism through which G-proteins mediate the enhancement of agonist activity [PMID: 27362234; 4726]. In contrast, ProtB, without G-protein interaction, potentially suffers from reduced ligand stability in its binding pocket, potentially leading to ligand release and rendering ProtB inactive. For ligands with limited binding affinities with APJR, or those as flexible as apelin-13, ProtAs might exert an inhibitory effect on the ligand binding and activation of ProtBs. Conversely, ligands with higher affinity (such as cmpd644, PDB ID: 7W0L, PMID: 35817871), or those with extensive interactions, (such as ELA, PDB ID: 7W0N, PMID: 35817871), seem to have minimal impact as they can still bind to ProtBs. While this hypothesis requires further exploration, it offers a plausible mechanism explaining how G-protein binding enhances agonist binding to the receptors (ProtAs) whereas ProtB may function as a negative allosteric modulator, reducing the overall signaling output when forming the dimer with ProtA.

- Page 10: "Suggesting a destabilization of dimerization leading to a transition from dimer to monomer upon agonist binding and G-protein coupling."

- Page 11: "The conformational rotation of Y105 by approximately 45 degrees towards TM2/3 and TM6/7 in this context plays a crucial role in triggering receptor activation (Fig. 5e, f)."

Authors should state whether the simulations show this rotation, too, or, at least, this tendency.

Response: We thank the reviewer for the comment. We did not conduct molecular dynamics (MD) simulations specifically for the antibody-bound structure, and the observation of Y105 rotation was solely based on structural findings. In the revised manuscript, we supplemented a figure to show the side chain conformation overlaid by cryo-EM map (see **Fig. R13**, also shown in the updated **Extended Data Fig. 7e, f**). We believe that the structural evidence is already sufficiently clear to support our findings. Meanwhile, we toned down the description by adding "The conformational rotation of Y105.....**might play a crucial role**....." (**line 251** in the revised manuscript).

Fig. R13 (also shown in the updated **Extended Data Fig. 7e, f**) | Cryo-EM maps of CDR3 region of JN241-9 in the orthosteric pocket, with the key residue Y105 highlighted: (a) CDR3 region of JN241-9 in the Gi protein-bound structure, shown as cartoon (left) and sticks (right), respectively; (b) CDR3 region of JN241-9 in the Gi protein-free structure, shown as cartoon (left) and sticks (right), respectively.

- Page 11: "meticulous analysis of the agonistic antibody"

What do authors mean with meticulous? Could you be specific? What type of analysis?

Response: We thank the reviewer for the comment. We have revised the analysis as “comprehensive structural analysis” in line 244 in our revised manuscript.

- Methods (MD): “A missing loop in one receptor was reconstructed by borrowing the corresponding symmetrical loop from a homologous receptor”

What loop? What homologous receptor?

Response: We thank the reviewer for the comment. We apologized that the description of this part is somewhat confusing. The cryo-EM map density for ProtB's intracellular loop 2 (ICL2) in the apelin-13-APJR-Gi complex is incomplete, so we did not model this part of ICL2 in the presented structure. To make this region complete for conducting the MD simulation, we grafted the symmetrical ICL2 from ProtA. At the same time, we made corresponding revisions to the manuscript in lines 752-754, highlighted in our revised manuscript as follows:

lines 752-754 (in Methods):

The missing intracellular loop 2 (ICL2) of ProtB in the $\text{dimAPJR}^{\text{AP13}}$ -Gi complex was reconstructed by utilizing the corresponding symmetrical ICL2 from ProtA of the same complex.

- Methods (MD):

The methodology for generating the replicas is not detailed in the methods. Are these replicas independent? Did authors build each replica independently in CHARMM-GUI? Otherwise, are they re-spawned from one system? This could be critical for the interpretation of results, specifically when one is interpreting the following 250 ns after a very short equilibration.

Response: We thank the reviewer for the comment. The entire process can be divided into three parts: minimization, equilibrium, and production. After we generate the system, we first carry out energy minimization. Then, using the results from the energy minimization, we independently repeat all subsequent procedures. That is to say, in these three trajectories, the step of energy minimization is the same, while the subsequent steps are all run completely independently. The corresponding description was supplemented in the Method Section in lines 768-770, also as follows:

lines 768-770 (in Methods):

The systems were initially subjected to a minimization process. Following this, leveraging the outcomes from the energy minimization, we independently reiterated all subsequent steps.

We extend our gratitude to the four reviewers for their diligent evaluation of our revised manuscript. Their constructive suggestions and insightful comments have contributed to the further improvement of this manuscript, as presented in this current version. Our point-by-point responses to each reviewer's new comments are listed below in blue text. The textural changes in the revised manuscript are highlighted in the marked-up version. All the line numbers (in red) indicated in this rebuttal are referred to the ones in the marked-up version.

Reviewer's Comments:

Reviewer #1 (Remarks to the Author):

Overall, authors have addressed proper responses on most of this reviewer's concerns in the revised text and the quality of the cryo-EM maps has been improved compared to the previous one. Nevertheless, I do have some comments on the overstated sentences which are added during the revision.

1. Line 185-190. "These structural changes suggest a potential scenario.."

I am still not convinced that the binding of an inactive protomer to the active protomer negatively regulates the activity of APJR in response to the specific biological response. There is no clear evidence for such negative allosteric effect. It could be simply interaction between the active and inactive protomers over the physiological concentrations. In my opinion, authors should mention such a possibility or should tone down their proposal:

Response: We thank the reviewer for the insightful comments. We understand the concern regarding the potential overstatement of the negative allosteric effect and the need for clear evidence to support such a claim. Given that the primary focus of our work is to provide solid structural evidence of the APJR dimer and explore the interplay of ligand, G proteins, and dimerization in regulating the APJR signaling, we agree that hypothesizing about negative allosteric modulation between the two protomers could be distracting. To address this concern, we have revised and toned down the discussion on this topic in the original section titled "Allosteric modulation in APJR homodimers through dynamic engagement of ProtB subunits". Meanwhile, some solid findings from this section have been moved to the next section, now titled "Apelin-13 bound APJR-Gi complexes revealed the co-existence of dimer and monomer." This combined section is now more focused and consistent in interpreting the structural findings. The overall conclusion on allosteric modulation between the two protomers has been simplified and summarized as: "...These findings reveal distinct ligand-binding behaviors within the APJR homodimeric structures." This revision can be found in lines 139-140 and is highlighted in the marked-up manuscript.

2. In discussion, Line 315-326: ".....These findings indicate that the binding of the Gi proteins may impose allosteric effect to the conformations at the dimer interface,

leading to a large-scale flexibility which in turn promotes the dissociation of the dimer...”

>> Some structures presented in this study are determined simply at low resolutions and it is difficult to reason that the low-resolution structures are related to the functional regulation of G-protein induced dissociation of the dimer. Authors have added this paragraph during the revision and no data support for this paragraph, which is clearly overstated.

Response: We thank the reviewer for this comment. We acknowledge that some of the structures were determined at relatively low resolution due to the intrinsic flexibility of the receptor complexes. However, we believe the structural data is solid enough to support the following conclusions: (1) In the absence of G proteins, the receptor predominantly exists in a dimer form, regardless of whether the ligand (agonist antibody) is bound or not. In addition, our previously published study in *NSMB* also demonstrated that in the absence of G proteins, the cryo-EM results almost show the APJR entirely in dimeric form (**Fig. R1**); (2) Upon addition of G proteins, we observe a shift toward monomers, especially in the presence of the agonistic antibody (which provides the highest stability compared to peptide or small molecule ligands). In this state, the receptor primarily exists in monomeric form when coupled to G proteins. Notably, the agonistic antibody allowed us to compare distinct structural features, such as oligomerization states, in the presence and absence of G proteins, presenting a key finding in our study. Therefore, we have organized this part of the study into a separate section—“**Structural investigation of APJR bound to agonistic antibody in the presence or absence of G proteins**”, to highlight the importance of the structural research on the agonistic antibody and its overall support to the main focus of this paper.

Fig. R1 | Representative cryo-EM 2D classification averages of APJR in the presence or absence of cmpd644 without coupling to Gi, from the previous work (PMID: 35817871) [REDACTED]

We maintain confidence in our proposal that G proteins may disrupt dimer stability, leading to dimer dissociation, which is essential for activation. However, we agree with the reviewer that our initial interpretation regarding the dimer interface and conformational changes was not sufficiently rigorous. To address this, we have removed the related claims, and ensure that our conclusions are fully aligned with the presented data.

3. Line 325-332: Authors presented their hypothetical working model (also in Ext Fig 9). This model could mislead the readers as if this is a general mechanism for the activation of APJR. In fact, only 6% increase of the APJR dimer has been observed in the presence of the small molecule agonist as shown in this manuscript. Moreover, the G-protein induced dimer to monomer conversion is observed only in specific agonist (JN241-9) and the APJR dimer-G complex is an active form in the presence of certain agonist. Thus, the authors should at least modify this paragraph to discuss the activation model in more conservative and careful manner.

Response: We appreciate the reviewer's thoughtful comments regarding the presentation of our hypothetical working model.

Firstly, we understand the concern about the potential misinterpretation of our proposed model as a general mechanism for APJR activation. As mentioned above, we still maintain confidence in our proposal that G proteins may disrupt dimer stability, leading to partial dimer dissociation, which is essential for activation. This summary is based on both previously published and current structures presented in this manuscript. This series of studies formed the basis for our hypothetical model. To summarize the structural findings, we added some texts in the discussion (lines 342-344 in the marked-up manuscript), stating that "...These findings provide new insights into the mechanistic role of G proteins in class A GPCR dimer dynamics and receptor activation."

However, we acknowledge the reviewer's concern that this model could be oversimplified or potentially misleading, and we agree that it may be more appropriate to remove the model and related discussion from the manuscript.

The single-molecule experiments in our study confirm that APJR dimers exist in living cells. Although the reviewer noted that the addition of small molecule agonists may not significantly affect the proportion of APJR dimers, we believe this proportion is influenced by a complex regulatory network involving ligands, G proteins, and other cellular factors, which cannot be fully resolved through single-molecule experiments alone. Our primary goal with these experiments was to confirm the presence of both monomeric and dimeric states of APJR in living cells, rather than to detail the precise regulatory roles of G proteins and ligands in the signaling system. Therefore, to explore the regulation of dimerization by ligands and G proteins further, we conducted a series of structural studies, ranging from the inhibited state to conditions with only ligands and finally to conditions with both ligands and G proteins. To clarify the transition from single-molecule experiment to the structural studies of the whole landscape of APJR complexes, which is the central goal of this study, we added some texts in the first part of the main text (lines 110-114 in the marked-up manuscript), stating that "APJR signaling likely operates through a versatile regulatory mechanism influenced by ligands, G-proteins as well as the oligomerization state—complexities that single-molecule experiments cannot fully resolve. Therefore, we proceeded with a series of structural investigations of APJR complexes in various ligand and G protein conditions to further dissect this regulation."

4. -In addition, for the structures presented, the cryo-EM maps for ICLs are some ECLs are not visible, onto which authors built the models. I think authors should list possible disordered regions in the text. Moreover, the side-chains of the Q chain of the dimAPJ_AP13 are overfitted:

Response: We have listed ICLs and ECLs regions as disordered region in the legend of **Fig. 1**. For the issue of overfitting in the side-chains of the Q chain of the dimAPJ_AP13 structure, we have also addressed, as stated in the legend of **Fig. 1** (see the highlighted text in the legend).

Reviewer #2 (Remarks to the Author):

The authors have addressed my concerns.

But I suggest clarifying the beginning of the title. Also, “APJR” should be replaced by “apelin receptor”.

Response: We sincerely thank the reviewer’s positive feedback and appreciate the reviewer’s additional suggestions. We have revised the title as follows:

Title: The Versatile Regulation of **Apelin Receptor Mediated by Dimerization, Ligand Binding, and G-Protein Coupling**

Reviewer #3 (Remarks to the Author):

The revised manuscript addresses some of the comments raised in the initial round of review, although the authors have largely chosen to do this through the removal of data rather than addressing technical comments with inclusion of additional or improved data. Specifically, the mutagenesis results were removed, rather than adding expression level controls. Do the mutants not actually express comparably? In the revised manuscript, essentially no functional data are presented, which makes it difficult to evaluate whether there is a meaningful biological insight here. The structural results are also somewhat poorly supported as was highlighted by other reviewers in the first round. Fig R5/Ex Data 7d show density for previously reported structures of APJ is poor, but this does little to increase confidence in the structures reported here. Fig R11/Ex Data 5g is similarly unclear, and likely compatible with other reasonable models of the Trp6.48 rotamer.

Overall, I believe the revised manuscript addresses some of the technical points raised by removing incompletely controlled data. The narrative is a bit clearer, although still somewhat difficult to follow (particularly in view of the lack of functional data). It is hard to conclude anything regarding biological importance of the results in the revised version of the manuscript.

Response: We thank the reviewer for the thoughtful feedback on our revised manuscript and further suggestions for improvement. We sincerely apologize for the removal of critical functional data in the last revision. This was done due to the release of the *Cell* paper on apelin-13 binding mode by Zhang Y et al. (PMID: 38428423) during our last revision, as we intended to avoid providing duplicating information. However, after thorough discussions with our collaborators, we believe that our functional data provide complementary and additional insights into apelin-13 and ELA binding to APJR. Therefore, we have now reintegrated the functional data into this version. This part has been mainly added to the section titled “**Molecular recognition of apelin-13 by APJR and comparison to ELA binding mode**” (lines 142-166).

We acknowledge the need to include mutagenesis data to provide meaningful biological insight, and ensure appropriate expression level controls to validate our findings. Therefore, we have supplemented our study with appropriate mutagenesis and functional data in **Fig. R2 (also shown in updated Fig. 2)**. Part of these functional data were included in the initial version, with necessary revisions to improve clarity. We think this data is essential in supporting the structural findings on the elucidation of apelin-13 binding mode and comparison to ELA, which is complementary to the findings reported by Zhang Y et al in their *Cell* paper (PMID: 38428423). Additionally, the corresponding extended data has been supplemented in **Extended Data Fig. 4**.

Moreover, we understand the reviewer’s concern regarding adding expression level controls. To address this, we have conducted additional experiments to verify that the

mutants express at levels comparable to the wild-type. The results were listed below. It indicated that the mutants exhibited almost similar expression levels with the wild-type (Fig. R3, also shown in updated Supplementary Table 1). However, it is worth noting that despite a 50% reduction in the expression level of the Y185A mutant, since our study focused on comparing the effects of the two ligands of apelin-13 and ELA, the results are still meaningful. The results demonstrated that, at equivalent expression levels, Y185A had a more significant impact on apelin-13 induced activation of APJR compared to ELA. We believe these additions will further enhance the biological relevance and robustness of our conclusions. Moreover, the first part of our manuscript (Investigating APJR dimerization dynamics at cell surfaces) also supported the biological significance of our study and emphasized the need of structural investigations. Combining ours and previous literature studies, numerous functional studies have indicated the dimerization of APJR; however, structural studies on this topic have been lacking. Therefore, we believe that our structural findings provide valuable insights with relevant biological implications. With the inclusion of additional functional data, we anticipate that our findings will offer greater biological significance.

Fig. R2 (also shown in updated Fig. 2) | Recognition mechanism of the endogenous apelin-13 by APJR and comparison to ELA binding mode. **a**, Binding pose of apelin-13. Apelin-13 is shown in orange sticks. ProtA^{AP13} is shown as blue cartoon. **b**, **c**, Key residues in the apelin-13 binding pocket in APJR. Apelin-13 residues are labeled in orange. Hydrophobic interactions with F13 of apelin-13 (**b**). Residues interacting with M11 of apelin-13 (**c**). **d**, Effects of key residue mutations on Gi-protein signaling in the apelin-13 binding pocket of APJR, measured by Glo-Sensor cAMP assay. Heatmap is generated on the basis of the $\Delta pEC50$ ($\Delta pEC50 = pEC50$ of mutant – $pEC50$ of WT APJR) for either ligand. Each column represents the data of an independent replicate. The corresponding data are shown in Supplementary Table 1. “ND” indicates no detectable signal.

Mutant	WT	F78A	I109A	M113A	Y185A	K268A	F291A	Y299A	Y271A
Expression (% WT)	100±0.00	122.07±13.23 ^{ns}	106.40±4.28 ^{ns}	84.50±4.41 ^{ns}	56.82±12.04*	124.10±16.95 ^{ns}	100.72±18.62 ^{ns}	88.52±1.98 ^{ns}	94.03±2.68 ^{ns}

Fig. R3 (also shown in updated Supplementary Table 1) | Expression levels of APJR-WT and various mutants. nsP > 0.05, *P < 0.05, **P < 0.01, ***P < 0.001 and ****P < 0.0001 by one-way ANOVA followed by Dunnett’s multiple comparisons test, compared with the response of the APJR-WT.

Additionally, we have put significant effort into refining the models during the first round of revisions, resulting in considerable improvements in the map density (**Fig. R4**, also shown in **Extended Data Fig. 3**). The resulting maps may still exhibit low-resolution feature at certain regions owing to the intrinsic flexibility of the APJR complexes at different functional states. Furthermore, we acknowledge that the descriptions corresponding to Ex Data 7d in our manuscript were somewhat inaccurate. Taking into account the feedback from the first reviewer as well, we have decided to remove Ex Data 7d.

Fig. R4 (also shown in **Extended Data Fig. 3**) | Cryo-EM density maps and models of representative helices from the $\text{dimAPJR}^{\text{AP13}}\text{-Gi}$, $\text{monAPJR}^{\text{AP13}}\text{-Gi}$, JN241-APJR, apo-APJR, JN241-9-APJR and JN241-9-APJR-Gi structures.

Furthermore, following the reviewer's suggestions, we have further refined the density map shown in the original Ex Data 5g (**Fig. R5**, also shown in updated Extended Data Fig. 6g). With the optimized map density, we propose that the side chain of W6.48 is more appropriately positioned in its current conformation.

Fig. R5 (also shown in updated Extended Data Fig. 6g) | Refined cryo-EM density maps for the residue W261^{6.48} within the JN241-APJR complex and $\text{dimAPJR}^{\text{AP13}}\text{-Gi}$ complex structures, respectively.

Finally, according to the reviewer's suggestion and the recommendation from Reviewer #1, we have toned down some of the discussions. We are committed to addressing the reviewer's concerns by providing additional and improved data. We believe that these efforts will significantly enhance the quality and impact of our study.

Reviewer #4 (Remarks to the Author):

Thanks for addressing all points. I only have one comments regarding the major point of my revision

- The authors decided to not extend the simulations or perform new trajectories, but to state that simulations do not support the increase in distance between protomers. Please make sure this is clear across the text, for example, please remove "supported by MD simulations" from the following sentence of the revised manuscript (page 12, lines 310-312):

"Subsequently, upon binding to G proteins, the distance between the two protomers of APJR increases, supported by MD simulations indicating dimer destabilization"

Response: We thank the reviewer for the positive evaluation of our efforts to address all the points the reviewer raised earlier.

We acknowledge the reviewer's concern regarding the consistent interpretation of the MD data. In the newly revised manuscript, we have removed the phrase "supported by MD simulations" from the sentence: "Subsequently, upon binding to G proteins, the distance between the two protomers of APJR increases (Fig. 4a), ~~supported by MD simulations~~ indicating dimer destabilization upon agonist binding and G protein coupling." in lines 337-339.

We extend our gratitude to all the reviewers for their diligent evaluation of our revised manuscript. We have carefully addressed the remaining concerns from Reviewer #3 and are pleased to present the final revised version in this submission.

Reviewer's Comments:

Reviewer #3 (Remarks to the Author):

The revised manuscript addresses some of the points raised. Data on expression levels are rather hidden, and should be emphasized more clearly in my view. I am still skeptical of the overall impact of this manuscript, although most of my major technical concerns are addressed. I hesitate about how important this insight is for a broad audience, especially in view of the numerous caveats.

Response:

We thank Reviewer #3 for the thoughtful feedback. To address concerns about the visibility of expression data, we have clarified this in the manuscript: "Mutations targeting surrounding residues ..., **at comparable expression levels ...**" and emphasized these data in **Supplementary Table 2** to ensure they are more prominently presented.

Regarding the manuscript's impact, we highlight that the apelin receptor is a significant drug target for heart failure, obesity, and muscle preservation, with growing relevance in the rapidly expanding obesity drug market. In this study, we present ligand binding modes, in relation to receptor dimerization, for the apelin peptide and two functional antibody molecules. Furthermore, our study addresses critical gaps in understanding class A GPCR dimerization by elucidating mechanisms outlined in our prior NSMB study (Yue et al., 2022) and demonstrating the pivotal role of G proteins in modulating receptor dimerization and activation. These findings not only advance drug development strategies for apelin receptor but also provide a robust methodology for studying GPCR dimerization and oligomerization, contributing to a broader understanding of GPCR biology.

We trust these clarifications address the reviewer's concerns and underscore the significance of our contributions.